# The metabolic enzyme fructose-1,6-bisphosphate aldolase acts as a transcriptional regulator in pathogenic *Francisella*

Jason Ziveri[1,2], Fabiola Tros[1,2], Ida Chiara Guerrera[1,3], Cerina Chhuon[1,3], Mathilde Audry[1,2], Marion Dupuis[1,2], Monique Barel[1,2], Sarantis Korniotis[1,4], Simon Fillatreau[1,4], Lara Gales[5], Edern Cahoreau[5] & Alain Charbit[1,2]

The enzyme fructose-bisphosphate aldolase occupies a central position in glycolysis and gluconeogenesis pathways. Beyond its housekeeping role in metabolism, fructose-bisphosphate aldolase has been involved in additional functions and is considered as a potential target for drug development against pathogenic bacteria. Here, we address the role of fructose-bisphosphate aldolase in the bacterial pathogen *Francisella novicida*. We demonstrate that fructose-bisphosphate aldolase is important for bacterial multiplication in macrophages in the presence of gluconeogenic substrates. In addition, we unravel a direct role of this metabolic enzyme in transcription regulation of genes *katG* and *rpoA*, encoding catalase and an RNA polymerase subunit, respectively. We propose a model in which fructose-bisphosphate aldolase participates in the control of host redox homeostasis and the inflammatory immune response.

[1] Université Paris Descartes, Sorbonne Paris Cité, Bâtiment Leriche, Paris 75993, France. [2] INSERM U1151 - CNRS UMR 8253, Institut Necker-Enfants Malades, Team 11: Pathogenesis of Systemic Infections, Paris 75993, France. [3] Plateforme Protéomique 3P5-Necker, Structure Fédérative de Recherche Necker and Université Paris Descartes, INSERM US24/CNRS UMS3633, Paris 75993, France. [4] INSERM U1151 - CNRS UMR 8253, Institut Necker-Enfants Malades, Team 16: Immunity in Health and Disease, Paris 75993, France. [5] Metatoul Platform, LISBP, Université de Toulouse, CNRS, INRA, INSA, Toulouse 31077, France. Correspondence and requests for materials should be addressed to A.C. (email: alain.charbit@inserm.fr)

Francisella tularensis is the causative agent of the zoonotic disease tularemia[1]. This Gram-negative facultative intracellular pathogen can infect humans by different means including direct contact with sick animals, inhalation, insect bites, or ingestion of contaminated water or food. F. tularensis is able to infect numerous cell types, including dendritic cells, neutrophils, macrophages as well as hepatocytes, or endothelial cells but is thought to replicate in vivo mainly in macrophages[2]. Four major subspecies of F. tularensis are currently listed: subsp tularensis (also designated Type A), subsp holarctica (also designated Type B), and F. tularensis subsp novicida[3]. These subspecies differ in virulence and geographical origin but all cause a fulminant disease in mice that is similar to tularemia in humans[4]. Although F. novicida is rarely pathogenic in humans, it is highly infectious in mice and its genome shares a high degree of nucleotide sequence conservation with the human pathogenic species. F. novicida is thus widely used as a model to study highly virulent subspecies[5–7].

Francisella virulence is tightly linked to its capacity to multiply in the cytosolic compartment of infected cells, and in particular of macrophages in vivo. The ability of Francisella to replicate within macrophages necessitates the coordinate control of three master transcription regulators, called MglA, SspA, and PigR[8, 9], which integrate the nutritional status of the pathogen to virulence gene expression in the host[10]. Francisella belongs to the limited group of intracellular bacteria, notably with Listeria monocytogenes and Shigella flexneri, that can gain access to -and proliferate within- the host cell cytosol[11]. Cytosolic bacterial multiplication often requires the utilization of multiple host-derived nutrients[12–15] and hexoses such as glucose are generally the preferred carbon and energy sources. In mammalian cells, glycolysis and the oxidative branch of the pentose-phosphate pathway occur in the cytosol, whereas the tricarboxylic acid (TCA) cycle, glutaminolysis, and β-oxidation take place in the mitochondria. In contrast, the anabolic reactions (gluconeogenesis and amino acid, nucleotide, and fatty acids biosynthesis) occur mainly in the cytosol. Hence, the bacterial enzymes responsible for glucose metabolism (catabolism and anabolism) are likely to play a key role in intracellular bacterial adaptation.

In the present study, we decided to address the role of the unique class II fructose-1,6-bisphosphate aldolase (FBA) of Francisella, a ubiquitous metabolic enzyme occupying a central position in glycolysis and gluconeogenesis pathways. Remarkably, FBA has been recently reported to play a role in the pathogenesis of two important human pathogens, highlighting the importance of metabolism in pathogenesis. In M. tuberculosis, FBA was shown to be required for growth in the acute phase and for survival in the chronic phase of mouse infections[16]. FBA was also shown to be essential for replication and virulence of the obligate intracellular parasite Toxoplasma gondii[17].

Two different classes of FBAs, with different catalytic mechanisms, have been described according to their amino acid sequences and designated Class I- and Class-II FBAs, respectively[18–20]. These aldolases have also been implicated in many "moonlighting" or non-catalytic functions, based upon their binding affinity for multiple other proteins, in both prokaryotic and eukaryotic organisms[21]. Class I FBAs are usually found in higher eukaryotic organisms (animals and plants). They utilize an active site lysine residue to stabilize a reaction intermediate via Schiff-base formation. Class I FBAs have been shown to interact with proteins displaying different functions predominantly involved in cellular structure, including notably F-actin, WASP, phospholipase D, and V-ATPase[22]. Class II FBAs are commonly found in bacteria, archae and lower eukaryotes, including fungi and some green algae. Some bacterial species, including Escherichia coli[23, 24], have been reported to express both types of the

enzyme. Class II FBAs have been further subdivided into type A and type B[25] on the basis of the amino acid sequence. Of note, type A enzymes have been found mostly involved in glycolysis and gluconeogenesis, while diverse metabolic roles and substrate specificities have been reported for type B aldolases[21].

Several attempts to disrupt the Class II fructose biphosphate aldolase genes from different bacterial species, including Escherichia coli, Bacillus subtilis, and Pseudomonas aeruginosa, have been unsuccessful, thereby suggesting that the Class II FBP aldolases were essential for the viability of these organisms. In M. tuberculosis, FBA was indeed demonstrated to be essential[16, 26].

Here we show that the gene fba is dispensable for Francisella survival and growth under defined conditions, and appears to play a regulatory role in pathogenesis. This enzyme appears to lie at the crossroad between carbon metabolism and the control of host redox homeostasis.

## Results

**Metabolomics reveal high glucose consumption in infected BMMs.** To assess the impact of bacterial infection and multiplication on the metabolism of infected macrophages, we first analyzed the metabolome of mouse bone marrow-derived macrophages (BMMs) and compared the concentration of a series of metabolites recorded in non-infected (NI) cells to those recorded in infected cells. We used for infection, either wild-type (WT) F. novicida (WT) or a F. novicida mutant with a deletion of the entire Francisella pathogenicity Island (designated ΔFPI mutant strain[27] unable to escape from phagosomes and, hence, to grow in macrophages. Values were recorded 1 and 24 h after infection (Fig. 1a). We analyzed, in parallel, the corresponding exometabolomes of these cells to get further insight into metabolite exchanges (consumption/production) between cells and medium (Fig. 1b).

The quantitative metabolomics analysis of intracellular metabolites was performed by ion chromatography and tandem mass spectrometry (IC-MS/MS)[28]. This approach allowed us to quantify central and intermediary metabolites from a number of metabolic pathways occurring in BMMs (i.e., glycolysis/ gluconeogenesis, pentose-phosphate pathway, TCA cycle, nucleotides biosynthesis, and activated sugars biosynthesis). Whereas at 1 h, overall only minor modifications were observed (Supplementary Fig. 2A), at 24 h, infection affected the central metabolite pools (Fig. 1 and Supplementary Fig. 2B). Remarkably, the concentrations of Glucose-6P, Glycerol-3P and P5P were higher in BMMs infected with WT F. novicida than the other two conditions (NI cells or cells infected with the ΔFPI mutant), suggesting that cytosolic multiplication of WT F. novicida could be responsible for their increased intracellular concentration (Fig. 1). Of note, a reduction of all the quantified metabolites of the TCA and of most of the metabolites from the Glycolysis/ Gluconeogenesis pathways was recorded with both the WT and the ΔFPI mutant (Fig. 1). The intracellular concentrations of fructose-1,6P, PEP, 2PG, and 3PG were also decreased, in both WT- and ΔFPI-infected BMMs, suggesting that the infection itself was sufficient to trigger these changes. Concomitantly, the concentrations of UDP-glucose and pentose-5P both increased in BMMs infected with WT F. novicida.

Of note, the changes observed in macrophages infected by Francisella may not strictly reflect the adaptation of the host cell metabolism and might correspond to the combined activity of bacterial and host metabolisms. However, the bacterial contribution to the metabolome should be -if not marginal- at least minor since: (i) approximately only 10% of macrophages cells are generally infected, in the infection conditions used; (ii) each infected cell generally do not contain more than a hundred

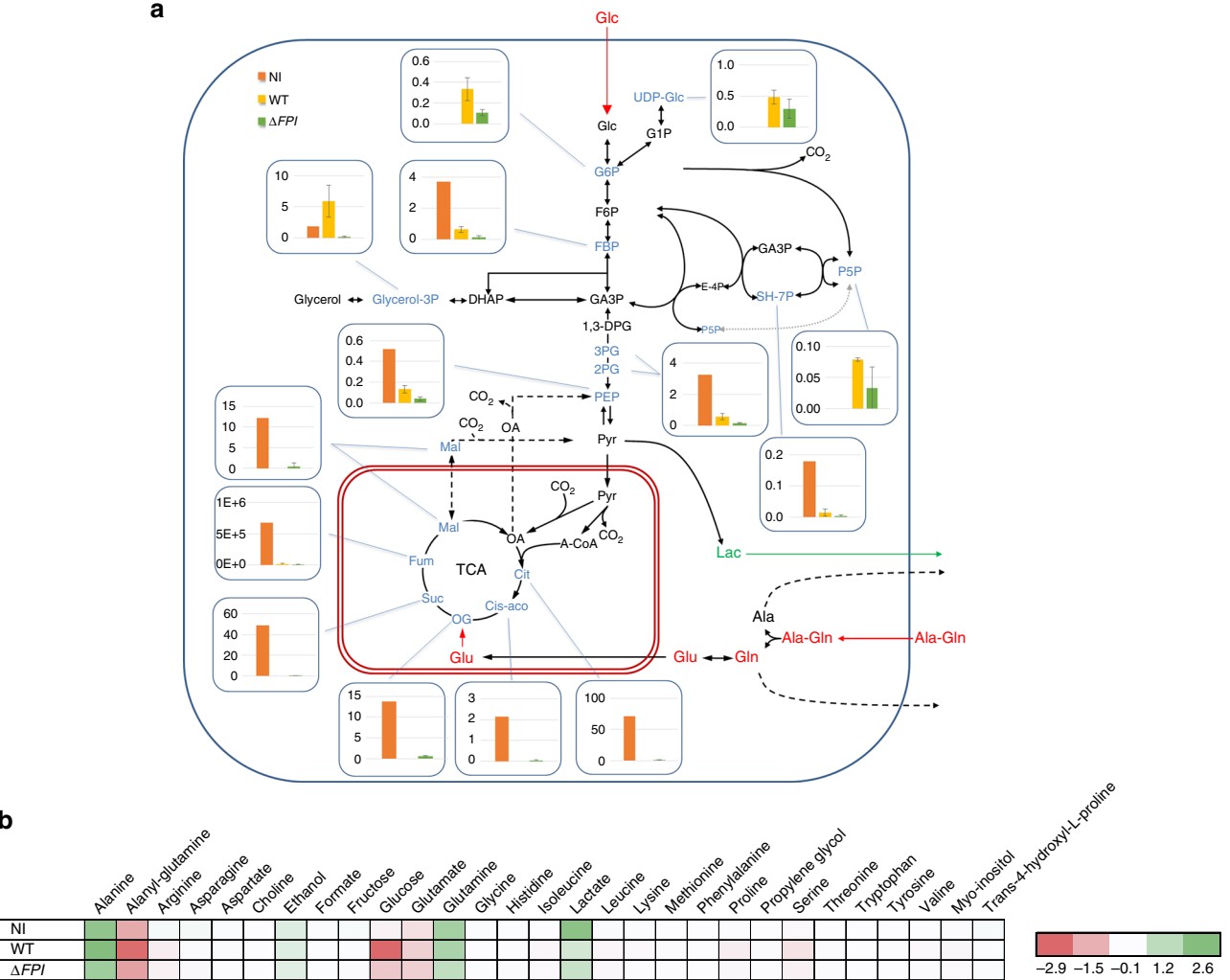

**Fig. 1** Carbon isotopologue distribution of central metabolites in BMMs. **a** Absolute concentrations of central metabolites in intracellular cell extracts (in μmol L$^{-1}$) after 24 h of cultivation of BMM macrophages: NI ($n=1$); or infected either with wild-type *F. novicida* (WT) ($n=3$) or ΔFPI (ΔFPI) strain ($n=3$). Fumarate measurements (Fum) represent MS peak area instead of absolute concentrations. **b** Consumed and produced metabolites, measured in extracellular media in the different infection conditions, profiled by NMR. Mean concentration changes, presented here by substracting concentration measured at 24 h to those measured at 1 h. Absolute concentrations changes are expressed in mM. *Positive/green values* correspond to metabolites accumulated in the media, *negative/red values* correspond to metabolites consumed by cells from the media between 1 and 24 h of incubation. Analyses were performed on biological triplicates and each sample was run in technical triplicates (mean and SD of metabolite concentrations were calculated using R 3.2.3, R Foundation for Statistical Computing, Vienna, Austria. URL http://www.R-project.org/)

bacteria; and (iii) the average volume of a bacterial cell is much lower than that of a macrophage, in the range of $1–2 \times 10^{-12}$ cm$^3$ per bacterium or $1–4 \times 10^{-9}$ cm$^3$ per macrophage cell[29]. Thus, it is reasonable to assume that the amount of metabolites contained in bacterial cells, in the samples analyzed, do not significantly contribute to the overall amounts measured.

We next performed the quantitative metabolomics analysis of the cell culture supernatants by nuclear magnetic resonance (NMR) to follow the evolution of the exometabolome during growth. This allowed us, by subtraction of concentrations measured at 24 and 1 h, to gain information on the substrates that were consumed from the medium and/or produced by the cells. The main substrates consumed in NI macrophages were alanyl-glutamine, glutamate, and glucose. In infected cells, the same substrates seemed to be consumed but a clear increase of glucose consumption was observed, especially for WT *F. novicida*. The main metabolites which accumulated in the media were alanine, glutamine, and lactate, in NI as well as in infected cells (Fig. 1b).

The metabolic modifications recorded in infected macrophages, and especially the glucose consumption, highlighted the importance of metabolic adaptation of *Francisella* for its intracellular survival and multiplication. We focused here on the role of the FBA in *Francisella* pathogenesis. This central enzyme of glycolysis and gluconeogenesis pathways (Supplementary Fig. 1) is at the crossroad of several other metabolic pathways that involve metabolites derived from glycolysis.

**FBA is required for growth on gluconeogenic substrates**. The FBA protein of *F. novicida* (FTN_1329) shows 99.2–99.4% amino acid sequence identity with its orthologues in other *F. tularensis* subspecies (Supplementary Fig. 3). FBA is also highly conserved in multiple other bacterial pathogens. For example, it shares 81.6% identity with *Pseudomonas aeruginosa* (PAO555) FBA, one of its closest homologs, and 74.3% with *Neisseria meningitidis* (NMA0587) FBA. However, it has only modest homology with FBAs of *Mycobacterium tuberculosis* (TBMG_00367) and *E. coli*

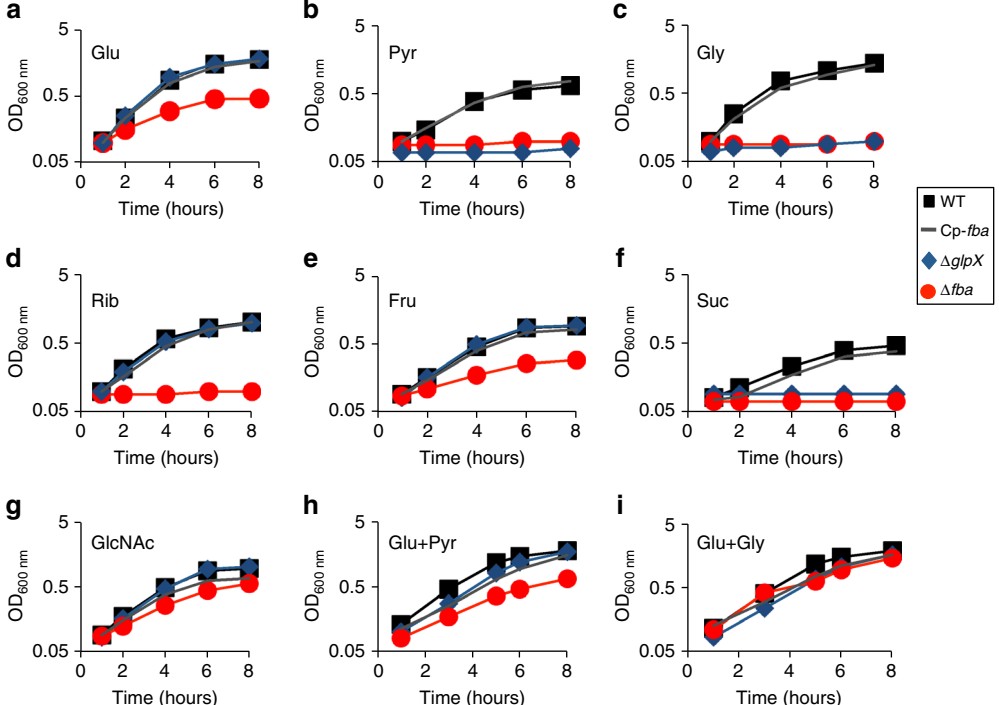

**Fig. 2** FBA inactivation inhibits growth of *Francisella* in the presence of various carbon sources. Wild-type *F. novicida* (WT, close *black squares*), isogenic Δ*fba* mutant (Δ*fba*, *red circles*), and complemented *fba* strain (Δ*fba* Cp-*fba*, *gray line*), and isogenic Δ*glpX* mutant (Δ*glpX*, *blue diamond*), were grown in chemically defined medium (CDM) lacking glucose and supplemented with different carbon source at a final concentration of 25 mM. **a** Glucose (Glc); **b** Pyruvate (Pyr); **c** Glycerol (Gly); **d** Ribose (Rib); **e** Fructose (Fru); **f** Succinate (Suc); **g** N-Acetylglucosamine (GlcNAc); **h** Glucose + Pyruvate (Glc + Pyr); **i** Glucose + Glycerol (Glc + Gly)

K12 (eco: b2925), with 27 and 25.6% amino acid identity, respectively. The FBA protein sequence does not contain any secretion-export signal and is predicted to be cytosolic. The *Francisella* proteins possess two signature motifs of FBA II according to Prosite database (PS00602 and PS00806) and bear a conserved C-terminal lysine like the FBA of *N. meningitidis*, suggesting that they are genuine IIB FBA (Supplementary Fig. 3).

We constructed an isogenic deletion mutant of *fba* in *F. novicida* by allelic replacement (Methods section) and first evaluated the impact of *fba* inactivation on bacterial growth in chemically defined medium (CDM[30], supplemented with various carbohydrates (Fig. 2). We included in these assays a Δ*glpX* mutant[31] lacking the strickly gluconeogenic enzyme fructose biphosphatase (FBPase; Supplementary Fig. 1). The Δ*fba* mutant (like the Δ*glpX* mutant) was unable to grow in media supplemented with gluconeogenic substrates such as pyruvate, glycerol, or succinate, a metabolite of the TCA cycle. In contrast, the Δ*fba* mutant showed only moderate growth defect in the presence of the glycolytic substrates glucose and fructose, as well as in the presence of N-acetylglucosamine (an amino sugar used for the synthesis of cell surface structures and entering the glycolytic pathway after its conversion into fructose-6P). WT growth was restored in the presence of both glucose and glycerol, confirming that FBA is required for growth of *F. novicida* in culture, mainly when gluconeogenic substrates are used as carbon sources.

It is likely that glycolytic substrates can use alternate route in the Δ*fba* mutant, and in particular the pentose-phosphate pathway (PPP). Indeed, we have recently demonstrated, using [13]C-labeled glucose, the recycling of carbons through the PPP in WT *F. novicida*. In contrast, when [13]C-labeled pyruvate was used, compounds of the PPP were not detected[31].

Of note, whereas the Δ*fba* mutant was unable to grow on ribose (a product of the PPP), the Δ*glpX* mutant showed WT growth on this sugar. WT growth was always restored in the Δ*fba*-complemented strain, confirming quantitative reverse transcription PCR (qRT-PCR) analyses, which demonstrated the absence of polar effect of the mutation (Supplementary Fig. 4). Since ribose cannot be converted into glucose-6P because the oxidative branch of the PPP is irreversible, FBA seems to be the only way for ribose to feed gluconeogenesis.

**fba inactivation impairs virulence**. We next evaluated the impact of *fba* inactivation in vitro, on intracellular multiplication; and in vivo, on virulence in the mouse model. The ability of wild-type *F. novicida* (WT) and Δ*fba* strains to survive and multiply in murine macrophage-like J774.1 cells was determined in cell culture medium supplemented either with glucose or glycerol or supplemented with both glucose and glycerol (Fig. 3a–c). We used a Δ*FPI* mutant strain as a negative control. In standard Dulbecco's modified eagle's medium (DMEM; i.e, supplemented with glucose), the intracellular multiplication of the Δ*fba* mutant was moderately affected (8–10-fold reduction of bacterial counts at 10 and 24 h, as compared to WT; Fig. 3a). In contrast, when glucose was substituted by glycerol, multiplication of the Δ*fba* mutant was severely impaired (Fig. 3b). The Δ*fba* mutant already showed a 10-fold reduction of intracellular bacteria compared to cells infected with the WT strain at 10 h; and a 1000-fold reduction of bacterial counts were recorded at 24 h (comparable to the Δ*FPI* mutant). Remarkably, when the medium was simultaneously supplemented with glucose and glycerol, the Δ*fba* mutant multiplied like the WT strain (Fig. 3c). Functional complementation (i.e, introduction of a plasmid-born WT *fba* allele into the Δ*fba* mutant strain) always restored WT growth.

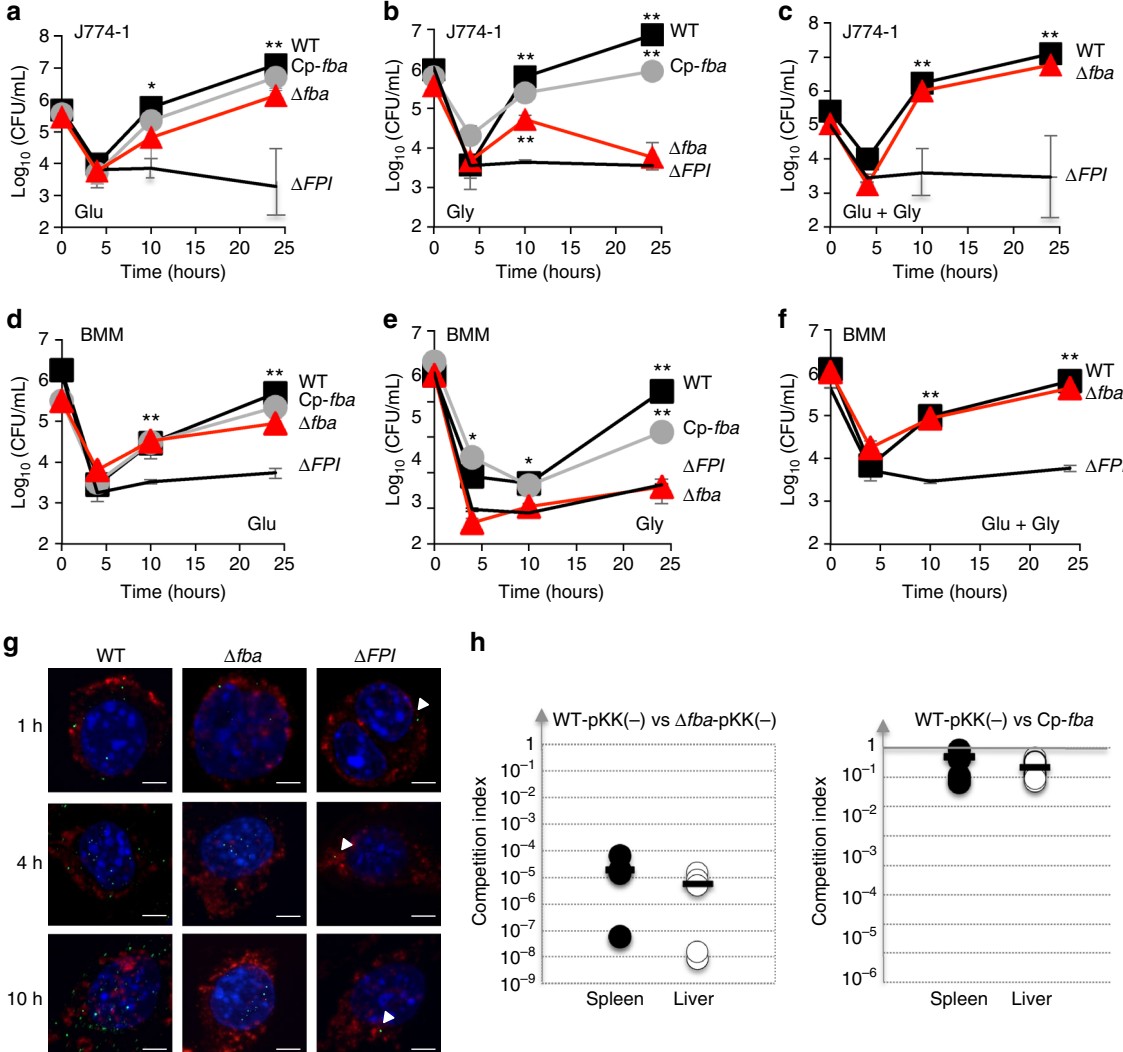

**Fig. 3** *fba* inactivation affects intracellular survival. Intracellular bacterial multiplication of wild-type *F. novicida* (WT, *black squares*), isogenic Δ*fba* mutant (Δ*FBA*, *red triangles*), and complemented Δ*fba* strain (Cp-*fba*, *gray circles*), and the Δ*FPI* negative control (*black lines*), was monitored during 24 h in J774A.1 macrophage cells and bone marrow-derived macrophages (BMM) from 6 to 8-week-old female BALB/c mice. DMEM (Dulbecco's modified eagle medium) was supplemented either with glucose **a**, **d**, glycerol **b**, **e**, or an equimolar concentration of glucose and glycerol **c**, **f**. **a–f** mean and SD of triplicate wells are shown. Each sugar was added to the cell culture medium at a final concentration of 5 mM. *$p < 0.01$; **$p < 0.001$ (determined by two-tailed unpaired Student's *t*-test). **g** Glycerol-grown J774.1 were infected for 30 min with wild-type *F. novicida* (WT), Δ*fba*, or Δ*FPI* strains and their co-localization with the phagosomal marker LAMP-1 was observed by confocal microscopy 1, 4, and 10 h, after beginning of the experiment. *Scale bars* at the *bottom right* of each panel correspond to 5 μM. J774.1 were stained for *F. tularensis* (*green*), LAMP-1 (*red*), and host DNA (*blue*, DAPI stained). **h** Group of five BALB/c mice were infected intraperitoneally with 100 CFU of wild-type *F. novicida* and 100 CFU of Δ*fba* mutant strain. Bacterial burden was realized in liver (*open circles*, *right column*) and spleen (*black circles*, *left column*) of mice. The data represent the competitive index (CI) value (in ordinate) for CFU of mutant/wild-type of each mouse, after 48 h infection, divided by CFU of mutant/wild-type in the inoculum. *Bars* represent the geometric mean CI value

The intracellular behavior of the Δ*fba* mutant was also tested in BMMs from BALB/c mice. Comforting the observations in J774-1 cells, growth of the Δ*fba* mutant was identical to that of the WT at 10 h and only slightly impaired (fivefold less bacterial counts) at 24 h (Fig. 3d) when BMMs were supplemented with glucose. In contrast, intracellular multiplication of the Δ*fba* mutant was severely impaired (Fig. 3e) when the culture medium was supplemented with glycerol. In medium supplemented with both glucose and glycerol, the multiplication defect of the Δ*fba* mutant was completely eliminated (Fig. 3f). Functional complementation restored normal intracellular bacterial replication in both cell types, in the presence of glucose. In the presence glycerol, complementation was fully restored up to 10 h and only partially at 24 h. Altogether these assays demonstrate that the Δ*fba* mutant is unable to multiply intracellularly when gluconeogenic substrates are used as carbon sources (Figs. 2–3). In contrast, when glycolytic substrates are used, the growth defect of the Δ*fba* mutant is essentially suppressed.

We confirmed that the presence of the empty vector (pKK214) in WT and Δ*fba* mutant strains used in these experiments had no impact on the phenotypes observed (Supplementary Fig. 5).

Interestingly, it has been recently shown that macrophages stimulated with the PPARβ/∂ agonist GW0742 increased their intracellular concentration of glucose[32]. This prompted us to test whether the addition of GW0742 to cell culture medium would result in increased intracellular bacterial multiplication of the Δ*fba* mutant. Addition of GW0742 had no visible effect on the multiplication of the Δ*fba* mutant in glycerol-grown J774-1 cells, during the first 10 h. However, a 15-fold increase in the number of intracellular Δ*fba* mutant bacteria was recorded at 24 h,

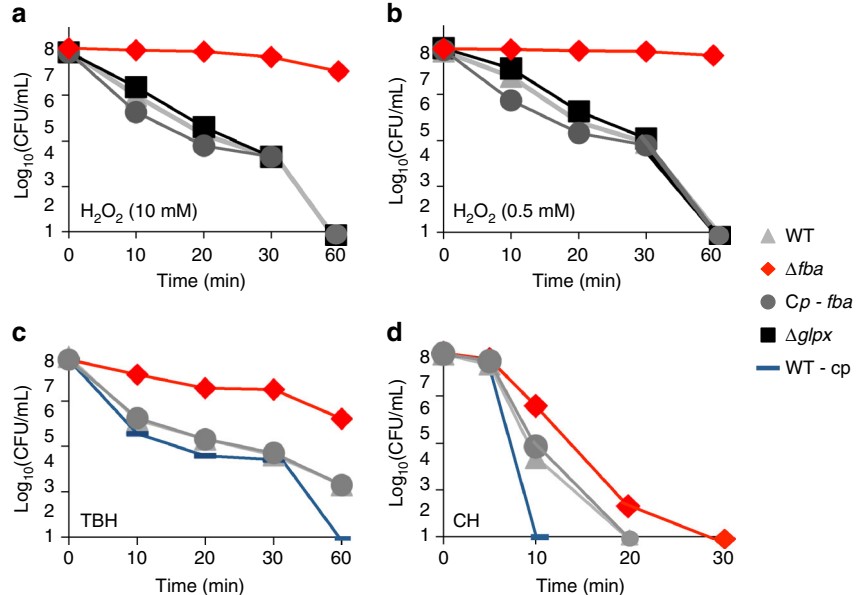

**Fig. 4** Oxidative stress survival. After an overnight culture in CDM supplemented with glucose and glycerol, bacteria were diluted in PBS and were subjected to oxidative stress **a** 10 mM $H_2O_2$, **b** 500 µM $H_2O_2$, **c** Tert-butyl hydroperoxide 10 mM (TBH), and **d** Cumene hydroperoxide 10 mM (CH). Bacteria were plated on chocolate agar plates at different time points and viable bacteria were numerated 2 days after. Experiments were realized three times

compatible with a GW0742-dependent stimulation of glucose production by these cells (Supplementary Fig. 6).

In order to determine whether the multiplication defect of the Δ*fba* mutant could be due to impaired phagosomal escape, we used confocal immunofluorescence microscopy. J774.1 macrophages were infected with either WT *F. novicida* or Δ*fba* mutant strain (the isogenic Δ*FPI* mutant was used as a negative control). Bacteria and the late phagosomal marker LAMP-1 were labeled with specific antibodies and their co-localization was monitored at three time points (1, 4, and 10 h). In glycerol-supplemented cells (Fig. 3g, Supplementary Fig. 7A), after 1 h, most of Δ*fba* mutant bacteria no were no longer associated with LAMP-1 (~13 and 21% of co-localization recorded with WT *F. novicida* and Δ*fba* mutant, respectively) and co-localization of the Δ*fba* mutant remained low after 4 h (~14%) and after 10 h (~15%). In glucose-grown cells (Supplementary Fig. 7B), after 1 h of infection, only 14 and 22% of co-localization was recorded with WT *F. novicida* and Δ*fba* mutant, respectively, indicating that the Δ*fba* mutant was able to escape phagosomes as fast as WT *F. novicida*. Co-localization of the Δ*fba* mutant remained still very low after 4 h (~10%) and after 10 h (~12%). In contrast, the Δ*FPI* mutant strain remained trapped into phagosomes, as illustrated by high co-localization with LAMP-1 at all time points tested (80, 85, and 88%, after 1, 4, and 10 h, respectively). Altogether, these results indicate that the intracellular growth defect of the Δ*fba* mutant, observed in glycerol-grown conditions, was not due to altered phagosomal escape but to impaired cytosolic multiplication.

Finally, to estimate the impact of *fba* inactivation on bacterial virulence, we performed an in vivo competition assay between WT and Δ*fba* mutant bacteria, in 7-week-old female BALB/c mice and monitored the bacterial burden in spleen and livers 2.5 days after infection by the ip route (Fig. 3h). The competition index recorded for the Δ*fba* mutant was very low in both target organs (between $10^{-3}$ and $10^{-4}$), demonstrating the importance of FBA in *Francisella* virulence in the mouse model.

**fba inactivation increases resistance to oxidative stress.** Upon entry into cells, *Francisella* transiently resides in a phagosomal

compartment that acquires bactericidal reactive oxygen species (ROS). We therefore examined the ability of WT *F. novicida* and Δ*fba* mutant strains to survive under oxidative stress conditions. We included in these assays a Δ*glpX* mutant as a control to evaluate the contribution of gluconeogenesis to oxidative stress. Bacteria were first grown in CDM supplemented with glucose + glycerol and then exposed to either to $H_2O_2$ (Fig. 4a, b), the organic hydroperoxides tert-butyl hydroperoxide (TBH, 10 mM) or cumene hydroperoxide (CH, 10 mM) (Fig. 4c, d) for 1 h. In the two $H_2O_2$ conditions tested (10 or 0.5 mM), the viability of both WT, Δ*glpX* and Δ*fba*-complemented strains was equally affected and behaved like the WT strain (Fig. 4a, b). In contrast, the Δ*fba* mutant was systematically more resistant to these stresses than the other strains. Indeed, after 30 min of exposure to $H_2O_2$, the viability of the Δ*fba* mutant was essentially unaffected whereas that of the other strains showed a 4 logs decrease in the number of viable bacteria. The Δ*fba* mutant showed also a greater resistance to organic hydroperoxides (CH and TBH), especially to TBH compared to WT. Of note, the FBA overproducing strain (WT strain bearing a plasmid carrying the WT *fba* gene) showed greater susceptibility to TBH and CH than WT. The fact that the Δ*glpX* mutant was as susceptible to oxidative stress as the WT strain was indicative of a role of FBA in oxidative stress resistance beyond its role in gluconeogenesis. We next examined the resistance of the mutant to other stresses: acidic, SDS, or upon incubation in the presence of 10% human serum. Under all the conditions tested, the viability of Δ*fba* mutant strain was undistinguishable from that of the parental strain (Supplementary Fig. 8).

The bactericidal activity of macrophages, and notably ROS production, is increased in the presence of IFNγ[33]. We therefore tested the intracellular multiplication of the Δ*fba* mutant compared to WT bacteria, in IFNγ-stimulated J774.1 macrophages. The Δ*fba* mutant showed improved survival and/or intracellular multiplication in the presence of IFNγ compared to WT (up to 20-fold increase in bacterial counts after 24 h), supporting the notion that *fba* inactivation confers to *Francisella*

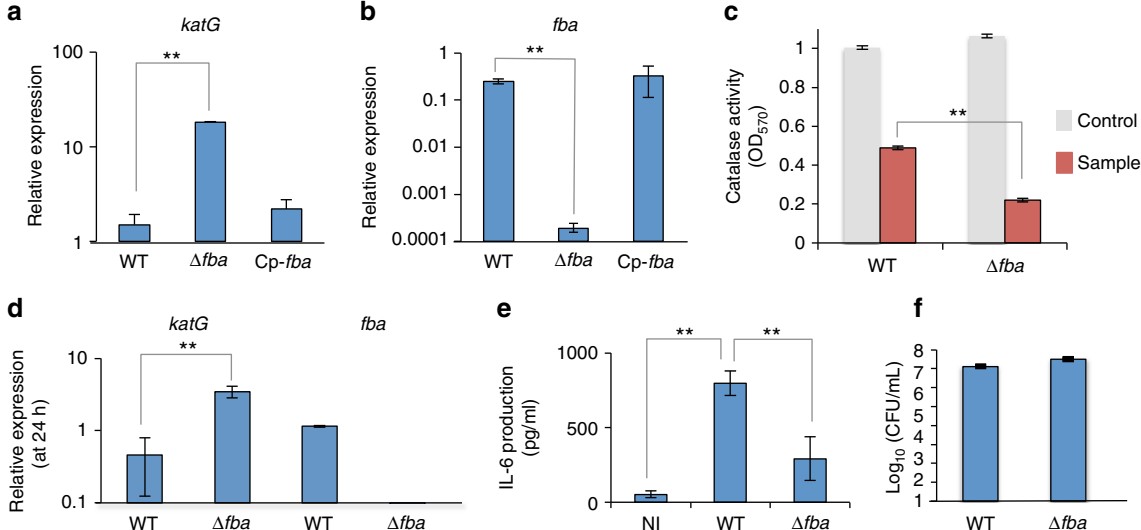

**Fig. 5** Quantitative real-time RT-PCR analysis. **a**, **b** Bacteria were grown overnight in TSB and qRT-PCR analyses were performed on selected genes, **a** *katG* gene or **b** *fba* gene in wild-type *F. novicida*, Δ*fba* mutant, and *fba*-complemented (Cp-*fba*) strains. For each gene, the transcripts were normalized to helicase rates (*FTN_1594*). **c** Catalase activity assay was realized at 570 nm, using the Catalase Assay Kit (ab83464; Abcam). The assay was performed according to manufacturer's recommendation. Each assay was performed on two independent protein lysates. The average of $OD_{570} \pm SD$ was recorded for wild-type *F. novicida* and the Δ*fba* mutant. The catalase activity recorded in the Δ*fba* mutant was significantly lower than that recorded in the wild-type strain (**$p < 0.01$). **d** J774.1 cells were infected for 24 h with wild-type *F. novicida* (*WT*) and Δ*fba* mutant. Total RNA was analyzed by quantitative RT-PCR with *katG* and *fba* gene. For each gene, the transcripts were normalized to helicase rates (*FTN_1594*) (**$p < 0.01$). **e** J774.1 cells were infected for 24 h with wild-type *F. novicida* (*WT*) and Δ*fba* mutant in DMEM supplemented with glucose and glycerol (25 mM). The supernatants were analyzed by ELISA to detect the amounts of IL-6 produced at 24 h in pg mL$^{-1}$. **f** Intracellular bacterial multiplication of wild-type *F. novicida* (WT) and Δ*fba* mutant was determined at 24 h, in the infected J774.1 cells used for the IL-6 dosage, as a control. (**$p < 0.01$). **a–c** mean and SD of three independent experiments are shown; **d–f** each assay was repeated at least three times. Mean and SD of three wells of one typical experiment are shown. *p*-values were determined by the two-tailed unpaired Student's *t*-test

an increased resistance to the ROS stress induced upon IFNγ treatment (Supplementary Fig. 9).

**A role for FBA in transcription**. The apparent specific impact of *fba* inactivation on oxidative stress resistance prompted us to first test the expression of the unique gene, *katG*, encoding the catalase responsible for the detoxification of $H_2O_2$ into $H_2O$ and $O_2$. The transcription of *katG* was quantified in WT and Δ*fba* mutant strains, in broth as well as in infected macrophages (Fig. 5). qRT-PCR analyses revealed that *fba* inactivation provoked a 12-fold increase of *katG* gene transcription in tryptic soya broth (TSB) (Fig. 5a) and complete inactivation of *fba* in the Δ*fba* mutant was confirmed (Fig. 5b). Catalase activity was also measured in whole-cell lysates from bacteria grown in the same TSB medium (Fig. 5c). Supporting the qRT-PCR data, increased catalase activity (twofold) was recorded in the Δ*fba* mutant compared to WT. We next quantified, by qRT-PCR, *katG* expression in infected J774-1 macrophages. We found that *fba* inactivation induced a sevenfold increase of *katG* transcription (Fig. 5d). As expected, we did not detect any *fba* transcript in macrophages infected with the Δ*fba* mutant.

It has been shown that a Δ*katG* mutant of *Francisella* provoked the rapid cytosolic accumulation of $H_2O_2$, triggering an inflammatory reaction via $Ca^{2+}$ signaling[34–36]. This *katG*-dependent control on the available intracellular $H_2O_2$ pool was proposed to control the production of pro-inflammatory cytokine and increased IL-6 production. We therefore thought to monitor the amount of IL-6 secreted in the supernatant of J774-1 macrophages, infected either with WT or with the Δ*fba* mutant (Fig. 5e). Supernatant from NI cells was used as negative control. As expected, IL-6 secretion, measured by ELISA[37], was significantly higher in the supernatant of WT-infected cells

compared to NI cells. Remarkably, IL-6 secretion was also significantly higher in the supernatant of WT-infected cells than in the supernatant of Δ*fba*-infected cells.

Intracellular bacterial counts, performed on the same cells as controls (Fig. 5f), confirmed that comparable numbers of intracellular bacteria were present in WT and Δ*fba* mutant at 24 h. These data are in agreement with a direct correlation between FBA-mediated *katG* repression and increased inflammatory response.

We next compared the amount of ROS in WT-infected cells to that in Δ*fba*-infected cells, 10 and 24 h after infection. For this, we used the 2′,7′-dichlorofluorescin diacetate (DCFDA) Cellular ROS Detection Assay Kit (Abcam, Cambridge, UK). The ROS content was ~20% lower with the Δ*fba* mutant compared to WT, at both time points (Supplementary Fig. 10A). As positive control, NI cells were stimulated with 5 µg mL$^{-1}$ of lipopolysaccharide from *E. coli* K12 (LPS-EK) Standard. LPS stimulation provoked an increase of ROS production that was up to 1.5-fold higher than that in WT-infected cells, at both time points tested. DCFDA levels were also visualized using fluorescence microscopy. The percentage of fluorescent cells was significantly higher in WT-infected cells (33%) or LPS-stimulated NI cells (63.5%) than in Δ*fba*-infected cells (8.5%) (Supplementary Fig. 10B). Altogether, these assays further supported a direct correlation between the FBA⁻mediated repression of KatG expression and cellular ROS production.

These data led us to hypothesize that FBA functions might extend beyond metabolic functions. To check whether *fba* inactivation could have a broader impact on protein expression than solely affecting KatG expression, we next performed a whole-cell comparative nanoLC-MS/MS proteomic analysis of WT, Δ*fba*, and FBA-overproducing (WT-cp), strains.

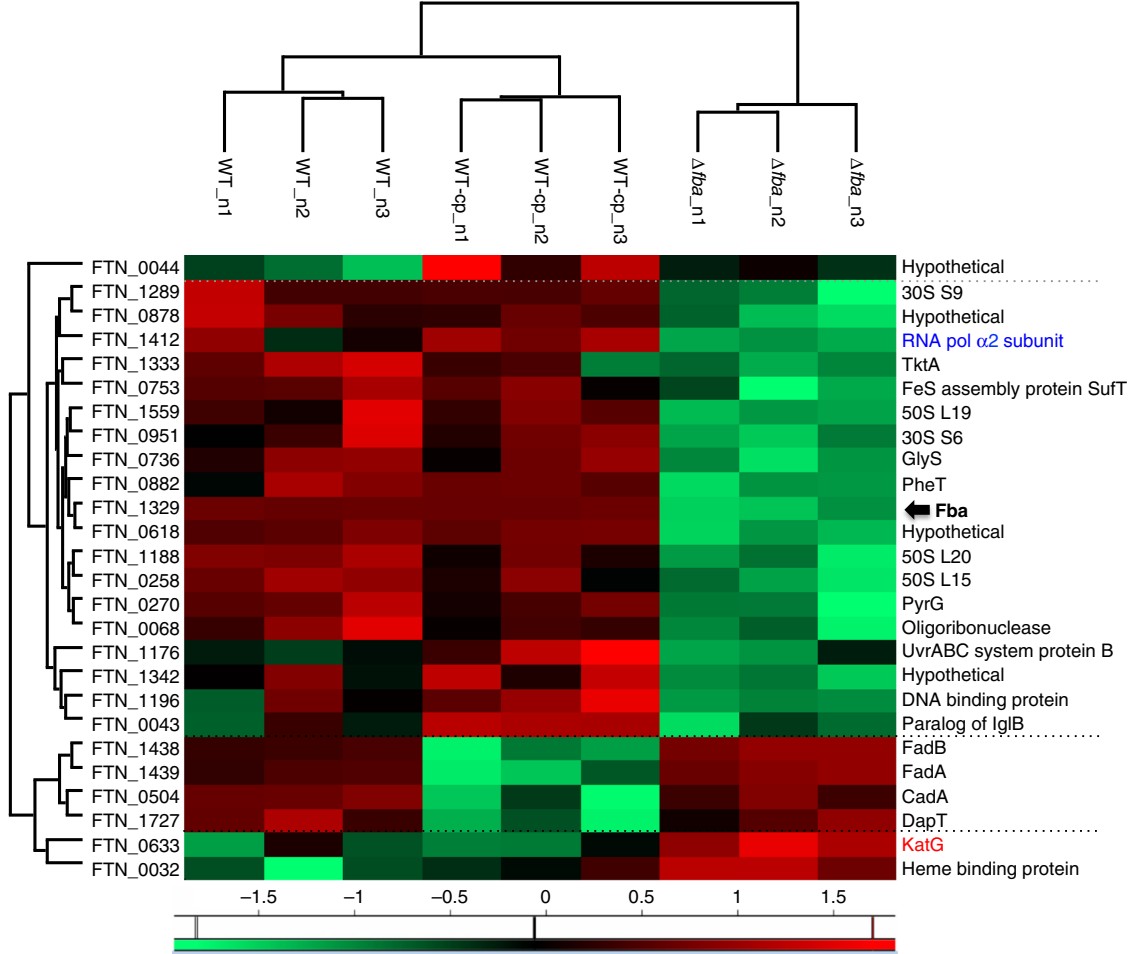

**Fig. 6** Biclustering and heat map of differential proteins. Heat map and biclustering were obtained by comparing the 26 differentially expressed proteins identified from label-free MS analysis of WT, Δ*fba*, and WT-complemented (WT-Cp) strains. To the *left*, FTN protein numbers; to the *right*, corresponding predicted functions. FBA, *in bold*, is indicated by a *black arrow*; RNA polymerase α2 subunit (down-regulated in Δ*fba*) is highlighted in *blue*, and KatG (up-regulated in Δ*fba*) in *red*

We could identify 1372 proteins across the 9 samples analyzed, which represent 80% of the predicted *F. tularensis* proteome. For statistical analysis, we kept 1154 proteins robustly quantified (data are available via ProteomeXchange with identifier PXD006908). An ANOVA test identified 26 proteins whose amount significantly differed between the three strains. Of note, most of them were conserved in the subspecies *holarctica* and *tularensis*. These 26 proteins were submitted to biclustering analysis and represented in the heatmap (Fig. 6). The column tree revealed that the protein profiles of the WT and the WT-cp strains were different from the Δ*fba* mutant.

Two opposite patterns could be distinguished: (i) a group of 19 proteins were expressed in lower amount only in the Δ*fba* mutant strain, suggesting that they are (directly or indirectly) positively regulated by FBA; and (ii) two proteins were in higher amount only in the Δ*fba* mutant strain, suggesting that they are (directly or indirectly) negatively regulated by FBA. The largest group of proteins under-represented or absent in the proteome of the Δ*fba* mutant strain comprised (in addition to FBA itself) proteins belonging to various functional categories, such as ribosomal proteins, a putative DNA binding protein and RpoA2, one of the two α subunits of RNA polymerase. Remarkably, fully supporting the transcriptional data, the catalase KatG was one of the two proteins expressed in higher amounts only in the Δ*fba* mutant strain (together with a putative Heme binding protein). Of note, the only down-regulated proteins in the FBA overproducing

strain comprised two proteins of the same operon (FadA and FadB) involved in fatty acid degradation, and two proteins linked to lysine metabolism. The fact that these pathways might also be affected by the over-production of FBA, suggest that FBA functions might extend beyond the proteins analyzed in this study.

In order to directly probe a role of FBA in transcription regulation, we constructed a His-tagged version of FBA, by fusing the His tag to its C-terminus (FBA-HA). The FBA-HA protein expressed in Δ*fba* was used to detect the in vivo binding of FBA to the promoters of genes corresponding to proteins whose expression varied over twofold in the proteomic comparison between WT and Δ*fba* mutant strains (Fig. 7). We performed chromatin immunoprecipitation (ChIP) followed by qRT-PCR analysis on two sets of genes: (i) two genes corresponding to proteins expressed in higher amounts in the Δ*fba* mutant (*katG* and *hemeBP*); and (ii) three genes corresponding to proteins expressed in lower amounts in the Δ*fba* mutant (*rpoA*, *uvrB*, and *fadA*) (Fig. 7a). We observed a nearly 30-fold enrichment of the *katG* promoter region and a fivefold enrichment of *hemeBP* promoter region. A nearly 10-fold enrichment of the *rpoA* promoter region was observed but no–or only very modest–enrichment was observed with the two other promoter regions (i.e., *uvrB* and *fadA*) (Fig. 7a). Transcription of *rpoA* was also significantly higher in the WT (>80-fold) and *fba*-complemented strains compared to the Δ*fba* mutant, suggesting

a direct role of FBA on *rpoA* transcription activation (Supplementary Fig. 11).

Transcription of seven additional genes, corresponding to proteins positively controlled by FBA, was also tested by qRT-PCR. Corroborating the proteomic analyses, transcription of these genes was higher in the WT and *fba*-complemented strains than in the Δ*fba* mutant. In particular, transcription of four of them (*pyrG*, *30S S9*, *glyS*, and *pheT*) was 50-fold to 200-fold higher in the WT strain than in the Δ*fba* mutant (Supplementary Fig. 11). We also monitored qRT-PCR, *katG* and *rpoA* gene expression in a Δ*glpX* mutant compared to WT *F. novicida*. Comparable expression of these two genes was recorded in the two strains, confirming that GlpX was not involved in their transcriptional control (Supplementary Fig. 11).

Electrophoretic mobility shift assays (EMSAs) were then performed to confirm direct binding of FBA to *katG* and *rpoA* promoter regions (Fig. 7b). For this, DIG-labeled double-stranded DNA fragments, corresponding to the regions immediately preceding each coding sequence (designated p*KatG* and p*rpoA*, respectively) were incubated in the presence of purified his-tagged FBA (FBA-HA) recombinant protein. With p*KatG*, a single band was detected with the labeled probe alone. Upon incubation with FBA-HA, a fraction of the probe was shifted and this shift was almost completely suppressed when a 125-fold excess of competing unlabeled specific p*KatG* oligonucleotide was added to the reaction (in addition to the labeled specific probe). With

p*rpoA*, two bands were detected with the labeled probe alone, possibly corresponding to different conformations of the probe. A major shifted additional band was detected upon incubation with FBA-HA as well as faint upper bands. The shifted bands disappeared when a 125-fold excess of competing unlabeled specific oligonucleotide was added to the reaction (Fig. 7b).

Finally, transcriptional p*katG-lacZ* and p*rpoA-lacZ* fusion were constructed and expressed in WT *F. novicida* or in Δ*fba* isogenic mutant strain (Fig. 7c). As expected, the β-galactosidase activity recorded with the p*katG-lacZ* construct was 3.5-fold higher in the Δ*fba* mutant background compared to WT. Conversely, the β-galactosidase activity recorded with the p*rpoA-lacZ* construct was 2.2-fold lower in the Δfba mutant background compared to WT.

Altogether, these assays confirmed the specificity of the interaction between FBA and *katG*, and *rpoA* promoter regions and supported that FBA specifically binds to different promoter regions, contributing either to transcriptional activation or repression.

## Discussion

Recent studies have shown that amino acids are likely to represent major carbon sources for intracellular *Francisella* and serve as gluconeogenic substrates[31, 38–44]. We show here that FBA is not involved in phagosomal escape but is critical for cytosolic multiplication in the presence of gluconeogenic substrates. Our metabolomics analysis performed on BMMs revealed that the infection by *Francisella* triggered numerous variations of intracellular metabolites, translating a complex metabolic response of the host to infection. Remarkably, a strong reduction of glucose present in the cell culture supernatant, with a concomitant increase of intracellular glucose-6P, was recorded after 24 h in infected BMMs, reflecting an increase of glucose consumption by

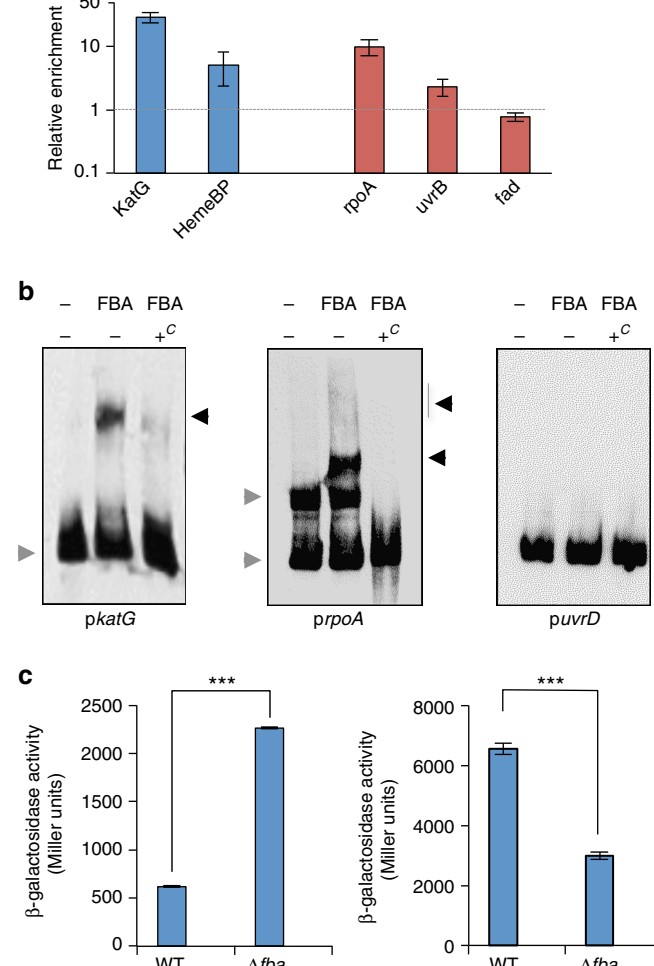

**Fig. 7** ChIP-qPCR, EMSA, and β-galactosidase assays. **a** ChIP-qPCR experiments were performed with Δ*fba* strain expressing a His-tagged version of FBA (Δ*fba*/cpFBA-HA) and with wild-type *F. novicida* (WT, negative control). The results are expressed as relative enrichment of the detected fragments. Mean and SD of three independent experiments are shown. Five promoter regions were tested: two (*blue* columns) corresponding to down-regulated proteins (according to the proteomic analysis) i.e., *katG* (KatG) and *hemeBP* (FTN_0032 or Heme Binding Protein); and three (*red* columns) to up-regulated proteins i.e., *rpoA* (RNA polymerase α subunit), *uvrB* (Endonuclease ABC subunit B), and *fadA* (Fatty acid degragation). **b** Electrophoretic mobility shift assays (EMSA) analysis of FBA—p*KatG* (*left*) and FBA—p*rpoA* (*right*) promoter interactions. EMSA assays were performed with DIG-labeled *katG* and *rpoA* promoter regions (p*katG*, 200 bp; p*rpoA*, 220 bp) and purified his-tagged FBA (FBA-HA) recombinant protein. Lane 1: labeled probe alone; lane 2, labeled probe incubated with 0.8 μg purified FBA-HA; lanes 3: probe incubated with 0.8 μg purified FBA-HA in the presence of 125-fold excess of corresponding unlabeled probe. The *gray arrows* (to the *left* of each panel) indicate the migration of the probe alone and the *black arrows* (to the *right*), the shifted bands. As negative control (*right panel*), EMSA assays were performed with DIG-labeled *uvrD* promoter region (p*uvrD*, 188 bp). Lane 1: labeled probe alone; lane 2, labeled probe incubated with 0.8 μg purified FBA-HA; lanes 3: probe incubated with 0.8 μg purified FBA-HA in the presence of 125-fold excess of corresponding unlabeled probe. As expected, no specific shifted bands were observed with this promoter region in presence of purified FBA-HA. **c** Quantification of *lacZ* expression in *F. novicida* wild type (WT) and Δ*fba* mutant strains containing either p*katG* (*left*) or p*rpoA* (*right*) promoter regions by β-galactosidase assay, as measured in Miller units. Each assay was repeated at least three times. Mean and SD of three wells of one typical experiment are shown. (*p*-values were determined by the two-tailed unpaired Student's *t*-test, ***p* < 0.0001)

these cells. In contrast to *L. monocytogenes*, which possesses a specific hexose phosphate transporter (Hpt)[45] required for virulence, *Francisella* cannot use the cytosolic pool of glucose-6P directly as a glycolytic substrate since its lacks a dedicated transporter. In this glucose-restricted context, a fully functional gluconeogenic pathway is thus likely to be critical for the bacterium to allow the utilization of alternative available nutrients (such as amino acids, glycerol, or other carbohydrates). The intracellular concentrations of all the measured TCA cycle metabolites decreased in infected BMMs. This could be due to their increased consumption by the cell in response to the infection and contributed by the multiplying bacteria themselves. In spite of that, the level of intracellular ATP appeared to have increased by at least twofold in WT-infected BMMs whereas it remained unchanged in Δ*FPI*-infected BMMs compared to NI macrophages. Other sources of ATP generation have been very recently described[46] which would account for the maintenance of a sufficient ATP pool in these cells. For comparison, infection with *L. monocytogenes* has been shown to increase glycolytic activity, enhance flux of pyruvate into the TCA cycle in infected BMM[47], and favor glucose uptake by infected cell and the production of compounds, such as glucose-6P, serine, and glycerol[48]. Altogether, these observations are compatible with the notion that cytosolic multiplying bacteria may take advantage of the glycolytic activity of the host to obtain a valuable source of metabolites for their own intracellular nutrition. The requirement of a functional FBA enzyme, for growth in the presence of gluconeogenic substrates and for virulence in the mouse, suggests that in vivo *Francisella* does not encounter the proper combination of carbon sources that could compensate for the lack of FBA.

*Francisella* produces several antioxidant enzymes such as the superoxide dismutases SodB and SodC, and the catalase KatG. In addition to these primary antioxidant enzymes, other proteins have also been shown to contribute (directly or indirectly) to oxidative stress defense, such as the alkyl-hydro-peroxide reductase AhpC, proteins with sequence similarities to the Organic hydroperoxide resistance protein (Ohr)[49] as well as the MoxR-like ATPase[38, 50]. The pleiotropic regulator OxyR is a primary regulator of oxidative stress in many bacteria[51] that responds to peroxides ($H_2O_2$). In *Francisella*, inactivation of *oxyR* confers high sensitivity to oxidants, deficient intramacrophage growth, and attenuated virulence in mice[52]. OxyR was also shown to bind to the upstream promoter regions of *katG*[53], indicating that *katG* expression is under the control of multiple regulatory mechanisms.

An earlier transcriptomic study, carried out in BMMs infected with *F. tularensis* Schu S4 strain, has shown that *katG* transcription was expressed during the first two hours after bacterial entry[42] and was later shut down for the rest of the infectious cycle whereas that of *fba* was induced after 4 h of infection and remained high up to 24 h. In agreement with this report, and further supporting the notion that FBA negatively regulates the expression of *katG* in infected cells, our qRT-PCR analysis (Fig. 5d) showed that *katG* expression was 7-fold up-regulated in the Δ*fba* mutant compared to WT, at 24 h of infection, in J774-1 macrophages supplemented with glucose and glycerol. These data suggest that the up-regulation of *katG* in the Δ*fba* mutant is not critical for intracellular multiplication in these conditions (in which the Δ*fba* metabolic defect is fully bypassed by appropriate carbohydrate supplementation). A role at later stages of the infectious cycle (such as cell to cell dissemination) and in other cell types is however possible.

ROS generated by macrophages upon infection have been shown to act as microbicidal effector molecules as well as secondary signaling messengers that regulate the expression of

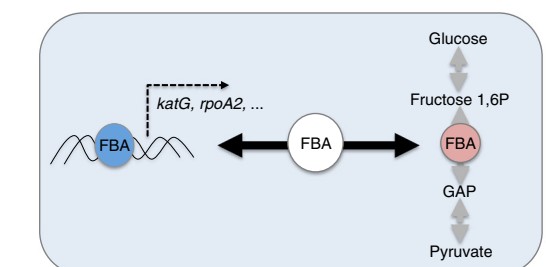

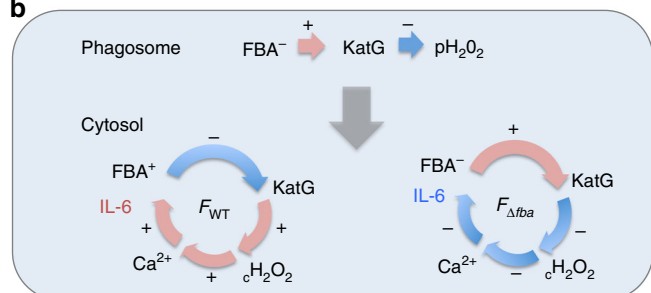

**Fig. 8** Proposed model of FBA regulation. **a** Schematic depiction of the dual role played by FBA: as a fructose biphosphate aldolase in glucose metabolism (*right*); and as transcription regulator (*left*). **b** Proposed *fba*-dependent regulation of *katG*. In the phagosomal compartment (*upper part*), *fba* is not expressed (FBA⁻), and *katG* expression is up-regulated (+) by $H_2O_2$ generated by phagosomal NADPH oxidase (p$H_2O_2$). In the cytosol (*lower part*), with the wild-type strain, *fba* expression is up-regulated (FBA⁺), which leads to *katG* down-regulation (−). Cytosolic $H_2O_2$ (c$H_2O_2$) then progressively accumulates which activates the $Ca^{2+}$ transporter TRPM2. Increased intracellular $Ca^{2+}$ ultimately activates inflammatory reactions and notably the production of IL-6. In contrast, with the Δ*fba* mutant (FBA⁻), *katG* expression is not repressed and cytosolic $H_2O_2$ is low. Consequently, the TRPM2-dependent $Ca^{2+}$ accumulation is limited and IL-6 production remains low

various inflammatory mediators. Of interest, the antioxidant action of the *katG*-encoded catalase of *Francisella* contributes to maintain the redox homeostasis in infected macrophages[34, 35]. The rapid cytosolic accumulation of $H_2O_2$, recorded in macrophages infected with a Δ*katG* mutant (i.e., lacking catalase activity), triggers a strong inflammatory reaction mediated by $Ca^{2+}$ entry via the transient receptor potential melastatin 2 (TRPM2)[36]. Indeed, TRPM2 channels are gated by $H_2O_2$ through cysteine oxidation[54–56].

The control of *katG* expression, mediated by FBA in the host cytosol, might constitute a mean of fine-tuning the redox status of the infected cell, leading to the control of pro-inflammatory cytokine and IL-6 production. Supporting this notion, we found that the total amount of ROS produced in cells infected with the Δ*fba* mutant (in which KatG expression is not repressed) was lower than in cells infected with the WT strain (Supplementary Fig. 10). Interestingly, previous studies have shown that *Francisella* also secreted KatG into the macrophage cytosol upon infection[57] as well as in the supernatant of bacterial cultures[58, 59]. Thus, KatG-dependent $H_2O_2$ detoxification might occur not only upon bacterial adsorption but might also be due to the direct action of the enzyme released in the host cytosol.

Altogether, these data are compatible with a sequential model of regulation (Fig. 8): (i) in the phagosomal compartment, *katG* expression is up-regulated due to the accumulation of $H_2O_2$ (designated p$H_2O_2$, derived from ROS produced by the NADPH oxidase) and FBA is not or poorly expressed; (ii) upon cytosolic escape, WT *Francisella* initially scavenges cytosolic $H_2O_2$, limiting

the activation of the calcium channel TRMP2 and delaying the $Ca^{2+}$ dependent inflammatory response[34]; (iii) the progressive increase of *fba* expression in actively multiplying bacteria leads to the concommittant reduction of *katG* expression; (iv) the accumulation of cytosolic $H_2O_2$ then stimulates the TRMP2-dependent entry of $Ca^{2+}$,[36]; and (v) ultimately leads to an increased production of IL-6, as described in this paper.

In contrast, in a Δ*fba* mutant strain, KatG production is not inhibited. In this context, the cytosolic concentration of $H_2O_2$ remains low and ultimately triggers only reduced IL-6 production in infected cells compared to WT bacteria.

Hence, FBA-mediated control of KatG expression should not be seen as a specific pathophysiologic role to render *Francisella* more vulnerable to a pro-inflammatory response but rather as a mean for the bacterium to combine metabolism and transcriptional regulation to optimally modulate the redox status of the infected cell. Indeed, one should bear in mind that intracellular survival and dissemination of the pathogen relies on its capacity to counteract both nutritional and innate immunity through an adaptation to the available nutritional resources and a tight and temporally-controlled dampening of cytokine production[60], ultimately leading to inflammasome activation and pyropsosis, allowing bacterial release and dissemination to adjacent cells.

Proteins that perform two or more distinct biological functions have been designated moonlighting proteins[61] and FBA can certainly be considered as a member of this family[21]. However, the only moonlight activity of FBA described thus far in pathogenic bacteria was a role as a secondary adhesin. For example, FBA is an essential enzyme in *M. tuberculosis* and also binds human plasminogen. Pneumococcal FBA is involved in the binding to human lung epithelial A549 cells via an interaction with a receptor belonging to the cadherin family. In *N. meningitidis*, FBA is not essential but is required for optimal adhesion to both human epithelial and endothelial cells[62, 63]. The recent detection of FBA in the cell wall fraction of *Coxiella burnetii*[64] and the identification of FBA orthologues in other pathogenic Gram-negative bacterial species, suggest that translocation of FBA to the Gram-negative cell envelope might constitute a more generalized phenomenon.

A putative role of metabolic enzymes in transcription regulation has been already suggested in bacteria, yeast as well as in eukaryotic cells. For example, in pathogenic *Streptococcus pyogenes*, the tagatose-1,6- bisphosphate aldolase LacD.1 acts as a negative regulator of the gene encoding the secreted cysteine protease SpeB[65]. This led to suggest that it might be a mechanism used by the bacterium to couple essential transcription programs to the sensing of its nutritional environment. In the yeast *Saccharomyces cerevisiae*, The class II FBA, in addition to its function in glycolysis, interacts physically with RNA polymerase III and plays a role in the control of its transcription[66]. In mammalian tissues, several enzymes involved in the glycolytic pathway display various non-glycolytic functions such as DNA repair and transcription, and are thus involved in important cellular functions including apoptosis, cell cycle control, and signaling pathways[67].

Our proteomic analyses showed that FBA exerted a specific repressive effect on only two proteins, including the catalase KatG. FBA appeared to be also involved in the up-regulation of multiple proteins including the α2 subunit of RNA polymerase (RNAP). Since the α subunit is a common site of interaction of RNAP with transcription activator proteins[68], affecting its expression is likely to alter transcription. Intriguingly, the *Francisella* genomes have the unique property to encode two different α subunits of RNAP with different regions predicted to be critical for dimer formation, promoter recognition, and activator interaction, suggesting possible distinct roles for the two α subunit in transcription regulation. Beyond this, our data also suggest that

FBA might be involved in other pathways such as fatty acid metabolism. The role of metabolic enzymes in the regulation of non-metabolic functions in pathogenic microorganism should deserve renewed attention.

## Methods

**Ethics statement**. All experimental procedures involving animals were conducted in accordance with guidelines established by the French and European regulations for the care and use of laboratory animals (Decree 87–848, 2001–464, 2001–486 and 2001–131, and European Directive 2010/63/UE) and approved by the INSERM Ethics Committee (Authorization Number: 75-906).

**Strains and culture conditions**. All strains used in this study are derivative from *F. tularensis* subsp. *novicida* U112 (*F. novicida* WT) and are described in Supplementary Table 1. Strains were grown at 37 °C on pre-made chocolate agar PolyViteX plates (BioMerieux), TSB or CDM supplemented with the appropriate carbon source at a final concentration of 25 mM. The CDM used for *F. tularensis* subsp *novicida* corresponds to standard CDM[30] without threonine and valine[39]. For growth condition determination, bacterial strains were inoculated in the appropriate medium at an initial $OD_{600}$ of 0.05 from an overnight culture in TSB.

**Stress assays**. Stationary-phase bacterial cultures were diluted at a final $OD_{600}$ of 0.1 in TSB broth. Exponential-phase bacterial cultures were diluted to a final concentration of $10^8$ bacteria per mL and subjected to either 500 μM $H_2O_2$, 10 mM $H_2O_2$, 10 mM Tertbutyl hydroperoxide, 10 mM Cumene hydroperoxide (1 h); pH 5.5 (1 h); 0.05% SDS (4 h); or 10% human serum (1 h). The number of viable bacteria was determined by plating apspropriate dilutions of bacterial cultures on Chocolate Polyvitex plates at the start of the experiment and after the indicated durations. Cultures (5 mL) were incubated at 37 °C with rotation (100 rpm) and aliquots were removed at indicated times, serially diluted and plated immediately. Bacteria were enumerated after 48 h incubation at 37 °C. Experiments were repeated independently at least twice and data represent the average of all experiments.

**Construction of chromosomal deletion mutants**. We inactivated the gene *fba* in *F. novicida* (*FTN_1329*) by allelic replacement resulting in the deletion of the entire gene (start and three last codons were conserved). We constructed a recombinant PCR product containing the upstream region of the gene *fba* (FBA-UP), a kanamycin resistance cassette (*nptII* gene fused with *pGro* promoter) and the downstream region of the gene *fba* (FBA-DN) by overlap PCR. Primers (Supplementary Table 2) *FBA* upstream FW (p1) and *FBA* upstream (spl_K7) RV (p2) amplified the 505 bp region upstream of position + 1 of the *FBA* coding sequence (FBA-UP), primers *pGro* FW (p3) and *nptII* RV (p4) amplified the 1091 bp kanamycin resistance cassette (*nptII* gene fused with *pGro* promoter); and primers *FBA* downstream (spl_K7) FW (p5) and *FBA* downstream RV (p6) amplified the 559 bp region downstream of the position + 1057 of the *FBA* gene coding sequence (FBA-DN). Primers p2 and p5 have an overlapping sequence of 12 and 12 nucleotides with primers p3 and p4, respectively, resulting in fusion of FBA-UP and FBA-DN with the cassette after crossing-over PCR. All single-fragment PCR reactions were realized using Phusion High-Fidelity DNA Polymerase (ThermoScientific) and PCR products were purified using NucleoSpin® Gel and PCR Clean-up kit (Macherey-Nagel). Overlap PCRs were realized using 100 ng of each purified PCR products and the resulting fragment of interest was purified from agarose gel. This fragment was then directly used to transform wild type *F. novicida* by electroporation[69]. Recombinant bacteria were isolated by spreading onto Chocolate agar plates containing kanamycin (5 μg mL$^{-1}$). The mutant strains were checked for loss of the wild type gene by PCR product direct sequencing (GATC-biotech) using appropriate primers.

**Functional complementation**. The plasmid used for complementation of the *F. novicida* Δ*fba* mutant (Δ*fba*), pKK-FBA$_{cp}$, is described below. Primers *pGro* FW and *pGro* RV amplified the 328 bp of the *pGro* promoter and primers *FBA* FW/*FBA* [PstI] RV amplified the 1064 bp *FBA* gene from U112. PCR products were purified and restricted using SmaI (*pGro* promoter) or PstI (*FAP*) restricted in presence of FastAP Thermosensitive Alkaline Phosphatase (ThermoScientific) to avoid self-ligation. A mixture of *pGro* promoter and interest gene fragments was then incubated in T4 Polynucleotide Kinase to allow blunt end ligation and fragments were then cloned in pKK214 vector after SmaI/PstI double restriction and transformed in *E. coli* TOP10. Recombinant plasmid pKK-FBA$_{cp}$ (designated Cp-*fba*) was purified and directly used for chemical transformation in *F. novicida* U112[38] by electroporation. Recombinant colonies were selected on Chocolate agar plates containing tetracycline (5 μg mL$^{-1}$) and kanamycin (5 μg mL$^{-1}$).

As controls, WT *F. novicida* and Δ*fba* mutant strains, carrying the empty vector pKK214, designated WT(−) and Δ*fba*(−), respectively, were also constructed and tested.

**Catalase assay**. Catalase enzyme activity was analyzed with a Catalase Assay Kit (Abcam, Cambridge, UK). WT *F. novicida* and Δ*fba* strains were grown overnight

in TSB supplemented with glucose and cysteine. $2 \times 10^6$ bacteria were harvested and used for the enzyme assay according to the manufacturer's instructions. After 30 min incubation time the reaction was stopped and the optical density was measured at $OD_{570}$ with a microplate reader. The assay was repeated twice, with similar results.

**Transcriptional analyses.** Isolation of total RNA and reverse transcription: For transcriptional analyses of bacteria grown in TSB, cultures were centrifuged for 2 min in a microcentrifuge at room temperature and the pellet was quickly resuspended in Trizol solution (Invitrogen, Carlsbad, CA, USA). For transcriptional analyses of bacteria in infected cells, J774-1 cells grown in standard DMEM were infected with either WT *F. novicida* (WT) or Δ*fba* mutant strain for 24 h. NI cells, incubated in the same conditons, were used a negative control. Cells were then collected by scratching, centrifuged at max speed in a microcentrifuge at room temperature and the pellet was quickly resuspended in Trizol solution.

Samples were either processed immediately or frozen and stored at −80 °C. Samples were treated with chloroform and the aqueous phase was used in the RNeasy Clean-up protocol (Qiagen, Valencia, CA, USA) with an on-column DNase digestion of 30 min. RNA RT-PCR experiments were carried out with 500 ng of RNA and 2 pmol of specific reverse primers. After denaturation at 70 °C for 5 min, 15 μL of the mixture containing 4 μL GoScript 5 × reaction buffer, 1 μL of 0.5 mM PCR Nucleotide Mix, 0.5 μL of RNasin Ribonuclease Inhibitor, and 1 μL GoScript Reverse Transcriptase (Promega) were added. Samples were incubated 5 min at 25 °C and then, 60 min at 42 °C, heated at 72 °C for 15 min and chilled on ice. Samples were stored at −20 °C. Different pairs of primers used in PCR to amplify the messenger RNA corresponding to the transcript of operon FTN_1333 to FTN_1329 (see Supplementary Table 2).

Quantitative real-time RT-PCR: WT *F. novicida* and mutant strains were grown overnight at 37 °C. Then, samples were harvested and RNA was isolated and reverse transcript to cDNA. The 25 μL reaction consisted of 5 μL cDNA template, 12.5 μL Fastart SYBR Green Master (Roche Diagnostics), and 2 μL 10 μM of each primer and 3.5 μL water. qRT-PCR was performed according to manufacturer's protocol on Applied Biosystems—ABI PRISM 7700 instrument (Applied Biosystems, Foster City, CA, USA). To calculate the amount of gene-specific transcript, a standard curve was plotted for each primer set using a series of diluted genomic DNA from WT *F. novicida*. The amounts of each transcript were normalized to helicase rates (*FTN_1594*).

**Chromatin immunoprecipitation-qPCR assay.** Chromatin immunoprecipitation (ChIP) was performed with wild-type *F. novicida* (WT) and a Δ*fba* strain expressing a His-tagged version of FBA (FBA-HA) (Δ*fba*/cpFBA-HA) bearing the 6x-His epitope fused at the C-terminal end of the protein). Bacteria were grown at 37 °C in 100 mL TSB supplemented to mid-log ($OD_{600}$ 0.3–0.4). Then bacteria were incubated in a final concentration of 1% formaldehyde (Sigma) for 30 min, after which glycine (Sigma) was added to a final concentration of 250 mM. Then, the bacteria were lysed by sonication (Branson Sonifier 250) to obtain a chromatin size of <500 pb. Aliquots were stored as input controls.

Immunoprecipitations were performed after an overnight incubation with the anti-HA antibody (6x-His Epitope Tag Antibody, Life Technologie), and with Dynabeads protein G (ThermoFisher Scientific). The beads were then washed and DNA was reverse-crosslinked and purified. Following ChIP, DNA was analyzed by qPCR. All ChIP-qPCRs were performed in triplicate from at least three independent experiments.

Briefly, the qPCR values were first normalized as follows: (i) qPCR values of the target promoter sequences (derived from ChIP and input samples) were divided by the qPCR values of the coding region of house keeping gene *uvrD* (Helicase) as internal control; (ii) the values obtained for *fba*/cpFBA-HA were next normalized by dividing them by their corresponding background values (derived from ChIP and input from WT). Then, the normalized signals from *fba*/cpFBA-HA derived from ChIP were divided by the normalized signals of *fba*/cpFBA-HA derived from input samples.

The results were expressed as relative enrichment of the detected fragments.

**Electrophoretic mobility shift assay.** We evaluated the ability of purified his-tagged FBA (FBA-HA) recombinant protein to bind to the *katG* or *rpoA* promoter regions (designated p*katG* and p*rpoA*, respectively), using EMSA. EMSAs were carried out using a DIG gel shift kit (Roche Diagnostic Corporation, Indianapolis, IN, USA), according to the manufacturer's instructions. Briefly, p*katG* (the 200 bp region immediately upstream of the ATG start codon of *katG*) was amplified with the pair of primers: Fw, 5′-GATATCGCTGGTGGATTATAAATAAATCG-3′ and Rv, 5′-GGTGATTTCCTCGCTATAAAGTTGA-3′; and p*rpoA* (the 220 bp region immediately upstream of the ATG start codon of *rpoA*) was amplified with the pair of primers: Fw, 5′-TCCAAACTCATATGTTATCCAGCAATAT-3′ and Rv, 5′-AGCAGTTTAAAACCTAGTTATATTTTATAG-3′), PCR products were resuspended in TEN buffer (10 mmol L$^{-1}$ Tris-HCl, 1 mmol L$^{-1}$ ethylene diamine tetraacetic acid (EDTA), 100 mmol L$^{-1}$ NaCl; pH 8.0) and labeled with DIG-ddUTP (Roche, Indianapolis, IN, USA) by the terminal transferase (Feng and Cronan, 2011). After 15 min of incubation of the DIG-labeled DNA probes (0.2 pmol) with 0.8 μg FBA-HA in binding buffer (Roche) at room temperature, the DNA-protein

complexes were separated using a native 5% polyacrylamide gel electrophoresis at 4 °C, transferred onto an equilibrated, positively charged nylon membrane (Amersham) followed by UV cross-linking (120 mJ for 180 s). Finally, the signals were captured by exposure to the high-performance chemiluminescence film (Amersham Hyperfilm ECL). As controls, we used the labeled probe without FBA-HA and the labeled probe incubated 30 min with the purified FBA-HA and with a 125-fold excess of unlabeled probe.

As a negative control, we used the promoter region of *uvrD*, a gene that is not regulated by FBA. p*uvrD* (corresponding to the 188 bp region immediately upstream of the ATG start codon of *uvrD*) was amplified with the pair of primers: Fw, 5′-TGCGACAAACTAATTTGTGAAACTTAG-3′ and Rv, 5′-CTGCCAGCACCAGCGAGA-3′. The labeled probe was incubated for 30 min: without FBA-HA, in the presence of FBA-HA, or in the presence of FBA-HA and with a 125-fold excess of unlabeled probe.

**β-galactosidase assays.** Reporter fusion construction: Two *lacZ* transcriptional fusion were constructed by cloning either p*katG* or p*rpoA* promoter region upstream of the *E. coli lacZ* gene. The p*katG* and p*rpoA* promoter regions were amplified using the primer pairs used in the EMSA and *E. coli s17-1 λpir* was the source of chromosomal DNA for amplification of the native *lacZ* gene. A 3075 bp region encompassing the entire coding sequence and its preceeding Shine-Dalgarno sequence was amplified with the pair of primers: Fw, 5′-TTAAT-TAAAGGAGGAACAGCTATG-3′ and Rv, 5′-TTATTTTTGACACCA-GACCAACTGG-3′). PCR amplifications were performed using Phusion High-Fidelity DNA Polymerase (ThermoScientific). The p*katG-LacZ* and p*rpoA2-LacZ* amplicons were then generated by overlap PCR. The purified products (3300 bp for p*katG-LacZ* and 3295 bp for p*rpoA2-LacZ*, respectively), flanked by SmaI and PstI sites were further digested with SmaI and PstI restriction enzymes, in presence of FastAP Thermosensitive Alkaline Phosphatase (ThermoScientific) to avoid self-ligation. Fragments were then cloned into pKK214 plasmid vector after SmaI/PstI double restriction and transformed into *E. coli* TOP10. The recombinant plasmids pKK-p*katG-LacZ* and pKK-p*rpoA2* were purified and directly used for cryo-transformation into *F. novicida* WT of Δ*fba* mutant strains. Recombinant colonies were selected on Chocolate agar plates containing tetracycline (5 μg mL$^{-1}$).

β-galactosidase assay: The assays were performed essentially as described previously[10]. Briefly, overnight cultures of reporter fusion construction strains were diluted in TS media and grown until mid exponential phase (0.2–0.8 $OD_{600}$). Bacteria were then harvested by centrifugation, resuspended in 1 mL of Z buffer with β-mercaptoethanol (60 mM Na$_2$HPO$_4$, 40 mM NaH$_2$PO$_4$, 10 mM KCl, 1 mM MgSO$_4$, pH 7.0, and 50 mM mercaptoethanol). SDS 0.2% and Chloroform were added and the samples were vigorously vortexed 15 sec and kept 5 min at room temperature. Then, 200 ul of ONPG (4 mg mL$^{-1}$ in Z Buffer) were added to the samples and incubated at 30 °C. The reactions were stopped by the addition of 1 M Na$_2$CO3. The absorbance at 420$_{nm}$ and 550$_{nm}$ was measured and the data converted in Miller units, using the classical formula $[(OD_{420} - 1.75 \times OD_{550})/(OD_{600} \times Volume \times Time)] \times 1000$.

**Cell culture and cell infection experiments.** J774A.1 (ATCC TIB-67) cells were propagated in DMEM (PAA), containing 10% fetal bovine serum (FBS, PAA) unless otherwise stated. BMM from 6 to 8-week-old female BALB/c mice were grown in Roswell Park Memorial Institute (RPMI-1640) or DMEM, containing 10% FBS. The day before infection, ~$2 \times 10^5$ eukaryotic cells (i.e., J774A.1 and BMM) per well were seeded in 12-well cell tissue plates (in appropriate cellular culture medium supplemented with the appropriate carbon source) and bacterial strains were grown overnight in 5 mL of TSB at 37 °C.

Infections were realized at a multiplicity of infection (MOI) of 100 and incubated for 1 h at 37 °C in culture medium supplemented with the appropriate carbon source (glucose, glycerol, or pyruvate). We used in our assays DMEM without glucose supplemented either with glucose at 5 mM or with other carbon sources at the same molarity. After 3 washes with cellular culture medium, plates were incubated for 4, 10, and 24 h in fresh medium supplemented with gentamycin (10 μg mL$^{-1}$). At each kinetic point, cells were washed 3 times with culture medium and lysed by addition of 1 mL of distilled water for 10 min at 4 °C. The titer of viable bacteria was determined by spreading preparations on chocolate plates. Each experiment was conducted at least twice in triplicates.

**IL-6 production.** Supernatants from J774-1 infected with either wild-type *F. novicida* (WT) or Δ*fba* mutant (MOI of 100) were harvested at 24 h. NI cells were tested as negative control. Cytokine were quantified by ELISA (BD) as previously described[37], using commercially available anti- IL-6 antibody in accordance with the manufacturer's instructions.

**ROS detection assay.** Intracellular ROS were detected by using the oxidation-sensitive fluorescent probe dye, DCFDA as recommended by the manufacturer (DCFDA Cellular ROS Detection Assay Kit, Abcam, Cambridge, UK). J774.1 cells were seeded at $4 \times 10^4$ cells per well. Cells were infected with bacteria for 10 or 24 h (MOI of 1000:1), washed three times with PBS and incubated with DCFDA diluted in PBS (15 μM). DCF fluorescence was measured with a multiplate reader Berthold TriStar (Berthold France SAS, Thoiry, France) with the use of excitation and

emission wavelengths of 480 and 525 nm, respectively. Values were normalized by protein concentration in each well (Bradford). Samples were tested in triplicates in two experiments.

**Determination of ROS generation via fluorescence microscopy**. J774.1 cells were seeded at $4 \times 10^4$ cells per well. Cells were infected with bacteria for 10 h (MOI of 1000:1), washed three times with PBS and incubated with DCFDA diluted in PBS for 1 h (15 μM). Images of the cells were captured with an Olympus CKX41 microscope and treated with Image J software. Cell counts were performed over 5 images of approx. 50 cells.

**Confocal experiments**. J774.1 macrophage cells were infected (MOI of 1,000) with wild-type *F. novicida* U112 (WT), the Δ*fba* isogenic mutant (Δ*fba*), or an isogenic strain deleted for the "*Francisella* Pathogenicity Island" (Δ*FPI*) in standard DMEM (DMEM-glucose) or DMEM without glucose and supplemented with either glucose or glycerol (5 mM each) for 30 min at 37 °C. Cells were then washed three times with PBS and maintained in fresh DMEM supplemented with gentamycin (10 μg mL$^{-1}$) until the end of the experiment. Three kinetic points (i.e., 1, 4, and 10 h) were sampled. For each point cells were washed with 1X PBS, fixed 15 min with 4% Paraformaldheyde, and incubated 10 min in 50 mM NH$_4$Cl in 1X PBS to quench free aldehydes. Cells were then blocked and permeabilized with PBS containing 0.1% saponin and 5% goat serum for 10 min at room temperature. Cells were then incubated for 30 min with anti-*F. novicida* mouse monoclonal antibody (1/500 final dilution, Creative Diagnostics) and anti-LAMP-1 rabbit polyclonal antibody (1/100 final dilution, ABCAM). After washing, cells were incubated for 30 min with Alexa488-conjugated goat anti mouse and Alexa546 conjugated donkey anti rabbit secondary antibodies (1/400 final dilution, AbCam). After washing, DAPI was added (1/1000 final dilution) for 1 min and glass coverslips were mounted in Mowiol (Cityfluor Ltd.). Cells were examined with an × 63 oil-immersion objective on a LeicaTSP SP5 confocal microscope. Co-localization tests were quantified by using Image J software; and mean numbers were calculated on more than 500 cells for each condition. Confocal microscopy analyses were performed at the Cell Imaging Facility (Faculté de Médecine Necker-Enfants Malades).

**Mouse infection**. WT *F. novicida* and Δ*fba* mutant strains were grown in TSB to exponential growth phase and diluted to the appropriate concentrations. Six to 8-week-old female BALB/c mice (Janvier, Le Genest St Isle, France) were intraperitoneally inoculated with 200 μl of bacterial suspension. The actual number of viable bacteria in the inoculum was determined by plating dilutions of the bacterial suspension on chocolate plates. For competitive infections, WT *F. novicida* and Δ*fba* mutant bacteria were mixed in 1:1 ratio and a total of 100 bacteria were used for infection of each of five mice. After 2 days, mice were killed. Homogenized spleen and liver tissue from the five mice in one experiment were mixed, diluted, and spread on to chocolate agar plates. Kanamycin selection to distinguish WT and mutant bacteria was performed. Competitive index (CI) [(mutant output/WT output)/(mutant input/WT input)]. Statistical analysis for CI experiments was as described[40] using the Student's unpaired *t*-test.

**Proteomic analyses**. Protein digestion: FASP (Filter-aided sample preparation) procedure for protein digestion was performed as previously described[70], using 30 kDa MWCO centrifugal filter units (Microcon, Millipore, Cat No MRCF0R030). Briefly, sodium dodecyl sulfate (SDS, 2% final) was added to 30 μg of each lysate to increase solubility of the proteins, in a final volume of 120 μL. Proteins were reduced with 0.1 M dithiotreitol (DTT) for 30 min at 60 °C, then applied to the filters, mixed with 200 μL of 8 M urea, 100 mM Tris-HCl pH 8.8 (UA buffer), and finally centrifuged for 15 min at 15,000 x *g*. In order to remove detergents and DTT, the filters were washed twice with 200 μl of UA buffer. Alkylation was carried out by incubation for 20 min in the dark with 50 mM iodoacetamide. Filters were then washed twice with 100 μl of UA buffer (15,000 x *g* for 15 min), followed by two washes with 100 μl of ABC buffer (15,000 x *g* for 10 min), to remove urea. All centrifugation steps were performed at room temperature. Finally, trypsin was added in 1:30 ratio and digestion was achieved by overnight incubation at 37 °C.

NanoLC-MS/MS protein identification and quantification: Samples were vacuum dried, and resuspended in 30 μL of 10% acetonitrile, 0.1% trifluoroacetic acid for LC-MS/MS. For each run, 1 μL was injected in a nanoRSLC-Q Exactive PLUS (RSLC Ultimate 3000, ThermoScientific, Waltham, MA, USA). Peptides were separated on a 50 cm reversed-phase liquid chromatographic column (Pepmap C18, Thermo Scienfitic). Chromatography solvents were (A) 0.1% formic acid in water, and (B) 80% acetonitrile, 0.08% formic acid. Peptides were eluted from the column with the following gradient 5 to 40% B (120 min), 40 to 80% (10 min). At 131 min, the gradient returned to 5% to re-equilibrate the column for 30 min before the next injection. Two blanks were run between triplicates to prevent sample carryover. Peptides eluting from the column were analyzed by data dependent MS/MS, using top-10 acquisition method. Briefly, the instrument settings were as follows: resolution was set to 70,000 for MS scans and 17,500 for the data dependent MS/MS scans in order to increase speed. The MS AGC target was set to $3 \times 10^6$ counts, while MS/MS AGC target was set to $1 \times 10^5$. The MS scan range was from 400 to 2000 *m/z*. MS and MS/MS scans were recorded in profile mode. Dynamic exclusion was set to 30 s duration. Three replicates of each sample were analyzed by nanoLC-MS/MS.

Data processing following nanoLC-MS/MS acquisition: The MS files were processed with the MaxQuant software version 1.5.3.30 and searched with Andromeda search engine against the NCBI *F. tularensis* subsp. *novicida* database (release 28-04-2014, 1719 entries). To search parent mass and fragment ions, we set a mass deviation of 3 and 20 ppm respectively. The minimum peptide length was set to 7 amino acids and strict specificity for trypsin cleavage was required, allowing up to two missed cleavage sites. Carbamidomethylation (Cys) was set as fixed modification, whereas oxidation (Met) and N-term acetylation were set as variable modifications. The false discovery rates at the protein and peptide level were set to 1%. Scores were calculated in MaxQuant as described previously[71]. The reverse and common contaminants hits were removed from MaxQuant output. Proteins were quantified according to the MaxQuant label-free algorithm using LFQ intensities[71, 72]; protein quantification was obtained using at least 2 peptides per protein.

Statistical and bioinformatic analysis, including heatmaps, profile plots, and clustering, were performed with Perseus software (version 1.5.0.31) freely available at www.perseus-framework.org[73]. For statistical comparison, we set three groups, each containing biological triplicate. Each sample was run in technical triplicates as well. We then filtered the data to keep only proteins with at least 2 valid values out of 3 in at least one group. Next, the data were imputed to fill missing data points by creating a Gaussian distribution of random numbers with a SD of 33% relative to the SD of the measured values and 1.8 SD downshift of the mean to simulate the distribution of low signal values. We performed an ANOVA test, $p < 0.01$, S0 = 1. Hierarchical clustering of proteins that survived the test was performed in Perseus on logarithmic scaled LFQ intensities after z-score normalization of the data, using Euclidean distances.

**Metabolomic analyses**. Central metabolite profiling by IC-MS/MS: Metabolome profiling of bone marrow-derived macrophages cells was performed in NI condition, in cells infected either with the wild-type *F. novicida* strain (WT) or with a Δ*FPI* isogenic mutant. Central metabolites of cells were harvested and quantified 1 and 24 h after infection. After elimination of cultivation medium by aspiration, adherent cells were washed with 3 ml of PBS buffer, also eliminated by aspiration. Cells were then rapidly quenched in cultivation plates with liquid nitrogen at −196 °C and extracted with 5 ml of a solvent mixture of ACN/Methanol/H$_2$O (2:2:1) at −20 °C. Samples were then evaporated and resuspended in 120 μL of ultrapure H$_2$O before analysis. The metabolites quantified were: Glucose-1-P (G1P), Glucose-6P (G6P), UDP-glucose (UDP-Glc) Fructose-6-P (F6P), Fructose-Bis-P (FBP), Pentoses-5-P = Ribose-5P + Ribulose-5P + Xylulose-5P (P5P), Ribose-1-P (R1P), Sedoheptulose-7-P (SH-7P), glycerol-3P (gly-3P), 2 and 3-PhosphoGlycerate (2/3-PG), Phospho-Enol-Pyruvate (PEP), Citrate (Cit), cis-aconitate (cis-aco), Oxoglutarate (OG), Succinate (Suc), Fumarate (Fum), Malate (Mal), Adenosine Mono, Di and Tri-phosphate (AMP, ADP, ATP), Cytidine Mono, Di and Tri-phosphate (CMP, CDP, CTP), Uridine Mono, Di and Tri-phosphate (UMP, UDP, UTP), Guanidine Mono and Di-phosphate (GMP, GDP), UDP-Acetylglucosamine (UDP-AcGluN), Shikimate-3-phosphate (Shikimate-3P), and Phosphor-Serine (P-Serine). Metabolite separation was performed by ionic chromatography (Dionex ICS 2500 ion chromatograph). Measurement of metabolite concentrations was performed by mass spectrometry with an Applied Biosystems 4000 Qtrap mass Spectrometer (ElectroSpray Ionisation in Negative mode; Detection mode: Multiple Reaction Monitoring). Quantification of metabolites from MS signals was made by external calibration with standards compounds mixture.

Exometabolome profiling by nuclear magnetic resonance: To estimate metabolites consumed and produced by the cells, samples of culture medium were harvested for each condition (NI, infected cells with WT and Δ*FPI F. novicida* strain) for each time point 1 and 24 h after infection. Extracellular samples were harvested by filtration of medium with a 0.45 μm diameter syringe filter. A 540 μL volume of extracellular sample was mixed with 40 μL Trimethyl-Sillyl-Propionic Acid 10 mM in D$_2$O. Measurement of exometabolome was performed by 1D 1H NMR with presaturation on a 500 Mhz NMR spectrometer equipped with a 5 mm TXI cryoprobe. NMR spectra were processed with Bruker Topspin 3.2, profiling and quantification of metabolites was performed with the Chenomx NMR suite 8.1. From spectra profiling, 29 metabolites (Supplementary Fig. 4) coming from original medium or produced by cells were unambiguously identified and quantified. We compared measured concentrations at 1 and at 24 h to evaluate substrates consumption in extracellular medium and metabolites production by cells.

**Data availability**. The mass spectrometry proteomics data have been deposited to the ProteomeXchange Consortium via the PRIDE partner repository with the dataset identifier PXD006908. The authors declare that all other data supporting the findings of this study are available within the paper and its Supplementary Information files.

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

## Acknowledgements

These studies were supported by INSERM, CNRS, and Université Paris Descartes Paris Cité Sorbonne. J.Z. was funded by a fellowship from the French "Délégation Générale à l'Armement" (DGA).

## Author contributions

J.Z. performed most of the in vitro experiments; F.T. performed murine in vivo experiments and some immunofluorescence experiments; I.C.G. and C.C. performed the proteomic analyses and I.C.G. analyzed and compiled the data; E.C. and L.G. performed metabolomic analyses and E.C. analyzed and compiled the data; M.A., M.D., and S.K. performed some in vitro experiments; M.B. analyzed the data; A.C. designed the experiments, and J.Z. and A.C. analyzed the data and wrote the paper. The funders had no role in study design, data collection and analysis, decision to publish, or preparation of the manuscript.

## Additional information

**Competing interests:** The authors declare no competing financial interests.

