## [Peer Review File · Nature Communications]

Reviewers' comments:

Reviewer #1 (Remarks to the Author):

- This paper characterizes the role of the evolutionary ubiquitous metabolic enzyme, fructose biphosphate aldolase (fba), in the pathogenicity of the intracellular bacterium, *Francisella novicium* (Fn). Using a genetic knockout of fba in Fn, the authors report several fascinating, if not unexpected, phenotypes. These include: (i) in vitro essentiality of fba for growth on gluconeogenic but not glycolytic carbon sources; (ii) a more pronounced attenuation of fba-deficient Fn in a macrophage-like cell line when cultured in presence of a gluconeogenic, but not glycolytic, carbon source; (iii) a profound attenuation of fba-deficient Fn in an in vivo mouse model of infection; (iv) a novel DNA binding transcriptional regulatory activity of fba, whose absence in fba-deficient Fn is associated with increased resistance to reactive oxygen intermediates, reduced IL-6 secretion, and increased survival in interferon-gamma activated macrophage-like cells.
- Based on these findings, the authors seek to ascribe Fn's pathogenicity, in part, to a combination of fba's canonical enzymatic activity and this novel non-metabolic transcriptional regulatory activity.
- Unfortunately, the work, as presented, is too incomplete to enable interpretation. For example, it remains entirely unaddressed why fba deficiency renders a specific in vitro phenotype with gluconeogenic over glycolytic substrates since fba is a shared enzyme of both pathways. In addition, it is entirely unclear why the fba mutant is sensitive to the carbon source provided in the macrophage cell culture medium (could it be that this phenotype reflects an extracellular, rather than intracellular event; can't tell from experimental methods). It is unclear how to interpret the metabolic profiling data since they reflect a combination of both host and microbial metabolites, and are pool size, rather than flux, measurements.
- Metabolic ambiguities aside, the pathophysiologic significance of fba's apparent transcriptional regulatory activity is entirely unclear as the metabolic defects seem enough to explain the observed attenuation in macrophages and mice. Moreover, the authors fail to identify a physiologic setting in which this transcriptional regulatory activity actually occurs (since genetic loss of fba itself seems unlikely in nature). Lacking this, it is difficult to imagine a specific pathophysiologic role for this activity as it would only seem to render Fn more vulnerable to the induction and effects of a pro-inflammatory/antimicrobial response. A more minor, but still important point, is that the authors fail to provide a direct demonstration of in vitro DNA binding and identification of specific operator sequences that could help to more rigorously identify target regulated genes.
- The section on inhibitor screening is uninformative and of unclear relevance in this manuscript.

Reviewer #2 (Remarks to the Author):

In their interesting study, Ziveri and colleagues show that the metabolic enzyme fructose-1,6-bisphosphate aldolase (FBA) is important for the intracellular replication of *Francisella tularensis*. They also present evidence that this same enzyme plays a role in controlling the expression of certain genes in this organism, which is particularly novel. The evidence that FBA is essential for intramacrophage growth when the growth medium contains gluconeogenic substrates is reasonably strong, although there may be an issue with the interpretation of some of the complementation data (see below). The evidence that FBA plays a regulatory role needs to be both stronger and more complete. Although the authors present reasonable evidence that FBA contributes to the control of redox homeostasis by controlling the expression of the *katG* gene, it's currently unclear whether the ability of FBA to control the expression of *katG* (or any other gene),

is important for intramacrophage growth or for virulence in mice.

Major comments

1. There may be a problem with the experiments in which the effects of the *fba* deletion are complemented with a plasmid-borne copy of the *fba* gene (Figs 2,3,4). From reading the materials and methods it looks as though the complemented *fba* mutant strain contains a plasmid that encodes *fba*. However, it's not clear whether the WT strain and *fba* mutant strains used in the same experiments contain an empty vector so that the results found with WT, *fba* mutant, and complemented strain can be interpreted properly. For these experiments it would of course be important for all cells to contain a plasmid vector.
2. The evidence that FBA plays a regulatory role needs to be stronger. The authors use proteomics to show that the abundance of ~30 proteins changes in an *fba* mutant compared to WT (Fig.6). However, the authors only present qRT-PCR evidence that FBA controls the expression of two genes (*katG* in Fig. 5 and *rpoA* in Fig. 7B). How many of the genes corresponding to the rest of the 30 proteins are controlled by FBA? A genomewide analysis (e.g. RNA-Seq) of the effects of FBA on gene expression would strengthen the study. If genomewide studies could not be performed, bolstering the findings of more extensive qRT-PCR analyses with the results of *lacZ* reporter assays would be suitable.
3. It would be important to show that the effects of *fba* on gene expression can be complemented with *fba* in trans (Figs 5 and 7).
4. The authors use ChIP followed by qPCR to show that an epitope-tagged version of FBA associates with a variety of promoter regions in *F. novicida*. However, it's not clear from the materials and methods whether these assays are truly quantitative. The authors should document how their assays are quantitative or use established procedures for quantifying enrichment by ChIP and qPCR.
5. Do the authors have any biochemical evidence that support direct binding of FBA to the *katG* or *rpoA2* promoter regions?
6. There is no test of whether the ability of FBA to control gene expression is important for pathogenesis. If FBA plays a key role in pathogenesis by modulating redox homeostasis through an effect on *katG* expression, is FBA important for virulence in cells that lack *katG*?
7. In relation to point 6. Don't the findings in Fig 5D and Fig 5F indicate that the ability of FBA to control the expression of *katG* (or other genes) is not important for intramacrophage growth?

Minor comments

8. Does a catalytically inactive variant of FBA still function as a regulator? I think this would be something that would be interesting to test, but not essential for the current study.
9. The y-axis on Fig. 7B should be labelled relative expression.

Reviewer #3 (Remarks to the Author):

This manuscript explores the role of a glycolytic enzyme, FBA, on *Francisella* virulence. The manuscript is clearly written, albeit heavily dependent on acronyms, and the experiments seem well-done. I was asked to comment on the proteomics work but I also have one general comment

about the key finding.

Major comments

1. The key conclusion here is that FBA is acting directly on transcription but I think the data does not quite fully support that, yet. Certainly the ChIP experiments are suggestive but it would be nice to see a negative control. For example, one possible explanation for the results is that an energy 'metabolon' is needed in the neighbourhood of those genes. If the same experiment were done with one of the up- or down-stream glycolysis enzymes, would one see the same enrichments as in Fig. 7?

2. Proteomics. I always get concerned when someone imputes data and then tries to calculate significance on those data since the imputation artificially suppresses the true biological variance. How many of the significantly changed proteins were those who had to have values imputed?

Minor comment:

1. In Fig. 1A, maybe indicate where the block in the pathway should be

Answers to reviewers

The comments and suggestions raised by the reviewers were carefully and thoroughly addressed. Clarifications were brought where needed and substantial additional experimental work was performed. The new data fully supported and consolidated our initial observations. Additional figures, text and references were included.

Reviewer 1

« This paper characterizes the role of the evolutionary ubiquitous metabolic enzyme, fructose bisphosphate aldolase (fba), in the pathogenicity of the intracellular bacterium, Francisella novicium (Fn). Using a genetic knockout of fba in Fn, the authors report several fascinating, if not unexpected, phenotypes. These include: (i) in vitro essentiality of fba for growth on gluconeogenic but not glycolytic carbon sources; (ii) a more pronounced attenuation of fba-deficient Fn in a macrophage-like cell line when cultured in presence of a gluconeogenic, but not glycolytic, carbon source; (iii) a profound attenuation of fba-deficient Fn in an in vivo mouse model of infection; (iv) a novel DNA binding transcriptional regulatory activity of fba, whose absence in fba-deficient Fn is associated with increased resistance to reactive oxygen intermediates, reduced IL-6 secretion, and increased survival in interferon-gamma activated macrophage-like cells. Based on these findings, the authors seek to ascribe Fn's pathogenicity, in part, to a combination of fba's canonical enzymatic activity and this novel non-metabolic transcriptional regulatory activity ».

• *“Unfortunately, the work, as presented, is too incomplete to enable interpretation. For example, it remains entirely unaddressed why fba deficiency renders a specific in vitro phenotype with gluconeogenic over glycolytic substrates since fba is a shared enzyme of both pathways. In addition, it is entirely unclear why the fba mutant is sensitive to the carbon source provided in the macrophage cell culture medium (could it be that this phenotype reflects an extracellular, rather than intracellular event; can't tell from experimental methods). It is unclear how to interpret the metabolic profiling data since they reflect a combination of both host and microbial metabolites, and are pool size, rather than flux, measurements”.*

Answer: It is likely that glycolytic substrates can use alternate route in the Δfba mutant, and in particular the pentose phosphate pathway (PPP). Indeed, we have recently demonstrated (Brissac *et al* 2015), using ^{13}C -labeled glucose in wild-type *F. novicida*, the recycling of carbons through the PPP. This information was added in the text (line 162-165).

Of note, in mammals, three forms of Class I FBAs are found (designated A, B or C). Aldolases A and C have been shown to be mainly involved in glycolysis, whereas aldolase B is involved in both glycolysis and gluconeogenesis. Hence, it is conceivable that class II FBAs may also display different roles in glycolysis or gluconeogenesis.

It is undeniably not possible to affirm that the changes observed in macrophages infected by *Francisella* strictly reflect the adaptation of the macrophage metabolism and we agree with the reviewer that it might correspond to the combined activity of bacterial and host metabolisms. However, we believe that the bacterial contribution to the metabolome should be -if not marginal- at least minor since: i) approximately only 10% of macrophages cells are generally infected, in the infection conditions used; ii) each infected cell generally does not contain more than a hundred bacteria after 24 h; and iii) the average volume of a bacterial cell is much lower than that of a macrophage (in the range of $1-2 \times 10^{-12} \text{ cm}^3$ per bacterium / $1-4 \times 10^{-9} \text{ cm}^3$ per cell). Thus, it is reasonable to assume that the amount of metabolites contained in bacterial cells, in the samples analyzed, do not significantly contribute to the overall amounts measured.

These precisions were added in the text: “... *The changes observed in macrophages infected by Francisella may not strictly reflect the adaptation of the host cell metabolism and might correspond to the combined activity of bacterial and host metabolisms. However, the bacterial contribution to the metabolome*

should be -if not marginal- at least minor since: i) approximately only 10% of macrophages cells are generally infected, in the infection conditions used; ii) each infected cell generally do not contain more than a hundred bacteria; and iii) the average volume of a bacterial cell is much lower than that of a macrophage (in the range of $1-2 \times 10^{-12} \text{ cm}^3$ per bacterium / $1-4 \times 10^{-9} \text{ cm}^3$ per cell; Krombach et al 1997). Thus, it is reasonable to assume that the amount of metabolites contained in bacterial cells, in the samples analyzed, do not significantly contribute to the overall amounts measured”.

There is currently no direct method to measure partitioning of metabolite pools in two types of cells in culture without prior cell separation, which is a difficult task to carry on, keeping a proper metabolism quenching (especially in the case of an intracellular infection). An experiment of ^{13}C isotopic profiling may highlight the active pathways that are actually operating during infection. However, this would necessitate to be in a metabolic stationary state, which is not the case in our experiments, since metabolic pools are evolving during infection. It would also most probably require a homogeneously infected cell population. Furthermore, even if ^{13}C flux analyzes could be used to measure metabolic fluxes, it would be an ambitious task to attribute flux partition between both organisms since same metabolic reactions can occur in parallel in the two organisms and give undifferentiated ^{13}C isotopic patterns from the same substrates

Despite these limitations, our analyzes allowed us to evaluate that metabolic pools were affected in infected macrophages compared to uninfected macrophages, and to identify the affected metabolic pathways. In particular, we clearly observed an impact of infection on TCA metabolites and on metabolites from gluconeogenesis/PPP above FBP (Glc6P, UDP-Glc, and Pentose-P metabolites – P5P and Sed7P).

• *“Metabolic ambiguities aside, the pathophysiologic significance of fba’s apparent transcriptional regulatory activity is entirely unclear as the metabolic defects seem enough to explain the observed attenuation in macrophages and mice. Moreover, the authors fail to identify a physiologic setting in which this transcriptional regulatory activity actually occurs (since genetic loss of fba itself seems unlikely in nature). Lacking this, it is difficult to imagine a specific pathophysiologic role for this activity as it would only seem to render Fn more vulnerable to the induction and effects of a pro-inflammatory/antimicrobial response. A more minor, but still important point, is that the authors fail to provide a direct demonstration of in vitro DNA binding and identification of specific operator sequences that could help to more rigorously identify target regulated genes”.*

Answer: The inability of the *fba* mutant to multiply in the presence of gluconeogenic substrates in cells reflects its inability to use these substrates as carbon sources (recapitulating the same phenotypes observed in broth). We have previously reported a comparable situation upon inactivation of the gluconeogenic enzyme GlpX (converting fructose 1,6-biphosphate to fructose 6-phosphate). Indeed, this mutant had a severe intracellular multiplication defect compared to wild-type, when cells were supplemented by gluconeogenic substrates (such as glycerol, pyruvate or amino acids; Brissac *et al.* 2015), but showed wild-type multiplication in medium containing glucose.

A somewhat comparable situation has been reported with the pathogenic bacterium *Legionella* (Price *et al* Science 2011), where the severe intracellular growth defect of an *ankB* null mutant could be rescued by supplementation with amino acids or pyruvate, as efficiently as by genetic complementation.

We think that the enzymatic and regulatory functions of FBA act synergistically and that their respective importance in virulence will depend on the cell type, the tissue and the phase of the infectious cycle in vivo. One should bear in mind that intracellular survival and dissemination of *Francisella* relies on a complex and tight temporally-controlled dampening of cytokine production. Indeed, during, active cytosolic multiplication, the bacterium transiently silences the AIM2 inflammasome but ultimately somehow takes advantage of caspase 1 activation and pyroptosis to promote its escape and dissemination to adjacent cells. These considerations were added in the Discussion « Hence, FBA-mediated control of *KatG* expression should not be seen as a specific pathophysiologic role to render *Francisella* more vulnerable to a pro-

inflammatory response but rather as a mean for the bacterium to combine metabolism and transcriptional regulation to optimally modulate the redox status of the infected cell. Indeed, one should bear in mind that intracellular survival and dissemination of the pathogen relies on its capacity to counteract both nutritional and innate immunity through: i) an adaptation to the available nutritional resources and ii) a tight and temporally-controlled dampening of cytokine production (Broz and Monack, 2011), ultimately leading to inflammasome activation and pyroptosis, allowing bacterial release and dissemination to adjacent cells».

Furthermore, FBA is not only acting on catalase expression but also regulates a number of other proteins and KatG itself is also under the control of additional regulatory stimuli. Thus, the action of FBA is integrated into a complex regulatory network during infection.

We have now confirmed the direct binding of FBA to the promoter regions of *katG* and *rpoA* genes by gel shift and β -galactosidase reporter assays. These results were added in the text and in Fig. 7.

- *“The section on inhibitor screening is uninformative and of unclear relevance in this manuscript.*

Answer: Following the reviewer’s suggestion, this section was removed.

Reviewer 2

“In their interesting study, Ziveri and colleagues show that the metabolic enzyme fructose-1,6-bisphosphate aldolase (FBA) is important for the intracellular replication of Francisella tularensis. They also present evidence that this same enzyme plays a role in controlling the expression of certain genes in this organism, which is particularly novel. The evidence that FBA is essential for intramacrophage growth when the growth medium contains gluconeogenic substrates is reasonably strong, although there may be an issue with the interpretation of some of the complementation data (see below). The evidence that FBA plays a regulatory role needs to be both stronger and more complete. Although the authors present reasonable evidence that FBA contributes to the control of redox homeostasis by controlling the expression of the katG gene, it’s currently unclear whether the ability of FBA to control the expression of katG (or any other gene), is important for intramacrophage growth or for virulence in mice”.

Major comments.

1. *“There may be a problem with the experiments in which the effects of the fba deletion are complemented with a plasmid-borne copy of the fba gene (Figs 2,3,4). From reading the materials and methods it looks as though the complemented fba mutant strain contains a plasmid that encodes fba. However, it’s not clear whether the WT strain and fba mutant strains used in the same experiments contain an empty vector so that the results found with WT, fba mutant, and complemented strain can be interpreted properly. For these experiments it would of course be important for all cells to contain a plasmid vector”.*

Answer: As requested by the reviewer, we have now compared wild-type *F. novicida*, containing -or not- the empty vector (pKK214), to Δfba containing -or not- the empty vector (pKK214).

We repeated growth experiments in CDM (CDM-Glucose, CDM-Glycerol and CDM-Glucose+Glycerol), and in cells (in J774-1 + Glucose and J774-1 + Glycerol). We also repeated the oxidative stress assay (+ H₂O₂ 1 mM). In each of the assays, the presence of the empty vector (pKK214) had no detectable impact on the properties of the strain (wild-type or Δfba mutant), fully confirming our previous results. The new data have been combined into a new Supplementary Figure (new **Fig S5**) and mentioned in the text.

2. *“The evidence that FBA plays a regulatory role needs to be stronger. The authors use*

proteomics to show that the abundance of ~30 proteins changes in an *fba* mutant compared to WT (Fig.6). However, the authors only present qRT-PCR evidence that FBA controls the expression of two genes (*katG* in Fig. 5 and *rpoA* in Fig. 7B). How many of the genes corresponding to the rest of the 30 proteins are controlled by FBA? A genomewide analysis (e.g. RNA-Seq) of the effects of FBA on gene expression would strengthen the study. If genomewide studies could not be performed, bolstering the findings of more extensive qRT-PCR analyses with the results of *lacZ* reporter assays would be suitable”.

Answer: As suggested, qRT-PCR analyses were performed on a series of additional genes corresponding to proteins controlled by FBA. In most cases, transcription of these genes was significantly higher in the wild type and *fba*-complemented strains compared to the Δfba mutant, corroborating the proteomic analyses (shown in new **Fig. S10**). The following paragraph was added: “Transcription of *rpoA* was also significantly higher in the wild-type (>80-fold) and *fba*-complemented strains compared to the Δfba mutant, suggesting a direct role of FBA on *rpoA* transcription activation (**Fig. S10**). Transcription of seven additional genes, corresponding to proteins positively controlled by FBA, was also tested by qRT-PCR. Corroborating the proteomic analyses, transcription of these genes was higher in the wild-type and *fba*-complemented strains than in the Δfba mutant. In particular, transcription of four of them (*pyrG*, *30S S9*, *glyS* and *pheT*) was 50-fold to 200-fold higher in the wild-type strain than in the Δfba mutant (**Fig. S10**)”.

LacZ reporter assays were also performed to further confirm the FBA-dependent transcriptional control of *katG* and *rpoA* genes. For this, transcriptional fusions were constructed between the promoter region of either *katG* or *rpoA* genes and the full length *lacZ* gene of *E. coli*. The resulting *p_{katG}-lacZ* and *p_{rpoA}-lacZ* transcriptional fusions were cloned into pKK shuttle vector and transferred into wild-type and Δfba mutant strains. β -galactosidase expression was 3.5-fold higher in the Δfba mutant compared to wild-type with the *p_{katG}-lacZ* reporter. Conversely, β -galactosidase expression was 2.2-fold lower in the Δfba mutant compared to wild-type with the *p_{rpoA}-lacZ* reporter. These new results, which fully supported the qRT-PCR and ChIP data, are described in the manuscript and included in **Fig. 7C**.

3. “It would be important to show that the effects of *fba* on gene expression can be complemented with *fba* in trans (Figs 5 and 7)”.

Answer: As requested, we have now shown that the effects of *fba* on gene expression could be fully complemented with *fba* in trans (strain Cp-*fba*), in vitro as well as in vivo. Accordingly, **Fig 5** (panels A, B) and **Fig 3** (panel H) were modified.

4. “The authors use ChIP followed by qPCR to show that an epitope-tagged version of FBA associates with a variety of promoter regions in *F. novicida*. However, it’s not clear from the materials and methods whether these assays are truly quantitative. The authors should document how their assays are quantitative or use established procedures for quantifying enrichment by ChIP and qPCR”.

Answer: As requested, we have documented in the Materials and Methods section the established procedures that we have applied for our quantifications.

The following sentences were added: “Briefly, the qPCR values were first normalized as follows: i) qPCR values of the target promoter sequences (derived from ChIP and input samples) were divided by the qPCR values of the coding region of house keeping gene *uvrD* (Helicase) as internal control; ii) the values obtained for *fba* /cpFBA-HA were next normalized by dividing them by their corresponding background values (derived from ChIP and input from WT). Then, the normalized signals from *fba* /cpFBA-HA derived from ChIP were divided by the normalized signals of *fba* /cpFBA-HA derived from input samples. The results are expressed as relative enrichment of the detected fragments”.

5. “Do the authors have any biochemical evidence that support direct binding of FBA to the *katG* or *rpoA2* promoter regions?”

Answer: We have now shown direct binding of FBA on the *katG* and *rpoA* promoter regions by gel shift assay. These results, which confirm the ChIP data, are included in the manuscript and shown in **Fig. 7B**.

6. “There is no test of whether the ability of FBA to control gene expression is important for pathogenesis. If FBA plays a key role in pathogenesis by modulating redox homeostasis through an effect on *katG* expression, is FBA important for virulence in cells that lack *katG*?”

Answer: The enzymatic activity and regulatory functions of FBA are likely to operate synergistically (see answers to reviewer 1). One may assume, from the phenotypes of *katG* mutants previously published (Lindgren et al; 2007; Guina et al. 2007), that a double mutant *fba-katG* might not necessarily show a more severe defect of intracellular multiplication than a single *fba* mutant, but should lead to an increased attenuation of virulence *in vivo*.

FBA should be important for virulence in cells that lack KatG because *fba* inactivation prevents the utilization of the available host-derived gluconeogenic substrates that are essential for its cytosolic multiplication (nutritional functions prevail).

7. “In relation to point 6. Don’t the findings in Fig 5D and Fig 5F indicate that the ability of FBA to control the expression of *katG* (or other genes) is not important for intramacrophage growth?”

Answer: The assays described in Figs 5D and 5F (performed in DMEM containing both glucose and glycerol) confirmed that *katG* expression was also controlled by FBA in cells. The reviewer is right, it is likely that *katG* up-regulation in the Δfba mutant is not critical for intracellular bacterial multiplication in these *in vitro* conditions (*i.e.* over a 24 h-period, in J774-1 cells) where the Δfba metabolic defect is bypassed by carbohydrate supplementation. A role at later stages of the infectious cycle (cell to cell spread and tissue dissemination) is however quite possible. The enzymatic and regulatory functions of FBA are probably not mutually exclusive. The relative importance of these functions in virulence is likely to depend on the phase of the infectious cycle, especially *in vivo*.

Accordingly, the following sentence was added: “These data suggest that the upregulation of *katG* in the Δfba mutant is not critical for intracellular multiplication in these conditions (in which the Δfba metabolic defect is fully bypassed by appropriate carbohydrate supplementation). A role at later stages of the infectious cycle (such as cell to cell dissemination) and in other cell types is however possible”.

Minor comments

8. “Does a catalytically inactive variant of FBA still function as a regulator? I think this would be something that would be interesting to test, but not essential for the current study”.

Answer: We agree with the reviewer that it would be something interesting to pursue in future studies. Conversely, catalytically active variants devoid of regulatory functions might also exist. However, one should bear in mind that it might not be possible to dissociate the two functions especially if structural links exist between them.

9. “The y-axis on Fig. 7B should be labeled relative expression”.

Answer: This was corrected.

Reviewer 3

“This manuscript explores the role of a glycolytic enzyme, FBA, on *Francisella* virulence. The manuscript is clearly written, albeit heavily dependent on acronyms, and the experiments seem well-done. I was asked to comment on the proteomics work but I also have one general comment about the key finding”.

Major comments

1. “The key conclusion here is that FBA is acting directly on transcription but I think the data does not quite fully support that, yet. Certainly the ChIP experiments are suggestive but it would be nice to see a negative control. For example, one possible explanation for the results is that an energy

'metabolon' is needed in the neighbourhood of those genes. If the same experiment were done with one of the up- or down-stream glycolysis enzymes, would one see the same enrichments as in Fig. 7?"

Answer: The reviewer wonders if one would see the same enrichments -or rather no enrichment- with other glycolytic/gluconeogenic enzymes.

Strongly playing against this notion, we show that a mutant lacking the gene *glpX* (encoding the gluconeogenic enzyme operating immediately upstream of FBA) was as sensitive to oxidative stress as the wild-type strain, implying that this enzyme is, in contrast to FBA, not acting on *katG* expression.

Furthermore, we have now also monitored, by qRT-PCR, *katG* and *rpoA* gene expression in a $\Delta glpX$ mutant compared to *F. novicida*. Comparable expression of these two genes was recorded in the two strains, confirming that GlpX is not involved in their transcriptional control. This information was added in the text and in **Fig. S10**.

Finally, we have confirmed direct binding of FBA onto *pkatG* and *prpoA* promoters by gel shift assay and *lacZ* reporter assays (Fig. 7B, 7C; see answer to reviewer 2).

2. "Proteomics. I always get concerned when someone imputes data and then tries to calculate significance on those data since the imputation artificially suppresses the true biological variance. How many of the significantly changed proteins were those who had to have values imputed?"

Answer: We agree with the reviewer that data imputation can suppress biological variance and it must be applied to a limited number of values. Out of the 26 significantly changing proteins, 7 had to have some missing values imputed. In terms of number of values concerned, out of the 234 values reported in the heatmap (26 proteins x 9 samples) less than 10% (23 values) were imputed. Imputation was mostly concerning proteins that were not detectable in the 3 samples of the same group (18 imputed values out of 23), suggesting that they were reproducibly absent or below the detection threshold, therefore very low in abundance.

Minor comment:

1. "In Fig. 1A, maybe indicate where the block in the pathway should be".

Answer: Fig. 1A corresponds to the metabolic pathway of the host cell (the bone marrow macrophage, infected or not). As mentioned in the text, at 24 h, a drastic reduction of all the quantified metabolites of the TCA and of most of the metabolites from the Glycolysis/Gluconeogenesis pathways was recorded with both the WT and the ΔFPI mutant. Of note, in Fig S1, which corresponds to the metabolic pathway of the bacterium, the place where FBA operates has been circled.

Reviewers' comments:

Reviewer #1 (Remarks to the Author):

In this revision, the authors have provided responses that, unfortunately, provide only indirect, if not oblique, responses to my previous critiques and leave key issues (critical to the impact of this work) unresolved. For example, they indirectly rationalize the selective essentiality of fba in gluconeogenesis over glycolysis by referencing previous work and inferring the basis for a selective growth defect on ribose not seen with a mutant in another gluconeogenic enzyme. However, they fail to provide direct experimental confirmation of this, which appears to be well within the authors' capabilities. This, in my opinion, is a key finding that would significantly strengthen/extend the impact of their work to clearly establish a metabolically specific/directional role for fba that is essential for virulence. In addition, the physiologic significance of the DNA binding/regulatory activity of fba continues to remain unclear with only speculative interpretation wanting in experimental support. Basically, it seems there is no experimentally identifiable role/phenotype for this activity (either in the ko where one would predict to see enhanced virulence and lower ROS levels; and, in the wild type situation, support accumulation of ROS and a pro-inflammatory response) as the ko is attenuated to a degree that can be simply explained by its canonical metabolic role. Lacking clear evidence for either a specific role in gluconeogenesis or altered ROS levels in infected cells (or perhaps even that fba levels are themselves regulated in any way), the significance of this work remains undefined.

Reviewer #2 (Remarks to the Author):

The authors have done a very nice job of addressing most of my concerns and they have significantly strengthened the evidence that FBA plays a regulatory role in Francisella. I have one additional comment relating to some of the new data they provide.

Comment

1. The experiments that show direct binding of FBA to the katG and rpoA promoters are a nice addition (Fig. 7B) but appear to lack a specificity control. This is important as it looks as though these electrophoretic mobility shift assays were performed in the absence of an excess of non-specific competitor DNA (which is normally included at a 100-1000-fold excess over the labelled probe). The authors should simply repeat their current assay using a labelled promoter that is not regulated by FBA as a negative control.

Reviewer #3 (Remarks to the Author):

The authors have addressed my concerns.

Answers to reviewers

The comments raised by the reviewers were addressed. The new data fully supported our observations. Additional figures and text were included.

Reviewer 1

In this revision, the authors have provided responses that, unfortunately, provide only indirect, if not oblique, responses to my previous critiques and leave key issues (critical to the impact of this work) unresolved. For example, they indirectly rationalize the selective essentiality of fba in gluconeogenesis over glycolysis by referencing previous work and inferring the basis for a selective growth defect on ribose not seen with a mutant in another gluconeogenic enzyme. However, they fail to provide direct experimental confirmation of this, which appears to be well within the authors capabilities. This, in my opinion, is a key finding that would significantly strengthen/extend the impact of their work to clearly establish a metabolically specific/directional role for fba that is essential for virulence. In addition, the physiologic significance of the DNA binding/regulatory activity of fba continues to remain unclear with only speculative interpretation wanting in experimental support. Basically, it seems there is no experimentally identifiable role/phenotype for this activity (either in the ko where one would predict to see enhanced virulence and lower ROS levels; and, in the wild type situation, support accumulation of ROS and a pro-inflammatory response) as the ko is attenuated to a degree that can be simply explained by its canonical metabolic role.

Lacking clear evidence for either a specific role in gluconeogenesis or altered ROS levels in infected cells (or perhaps even that fba levels are themselves regulated in any way), the significance of this work remains undefined.

Answer: As requested by the reviewer, we quantified ROS levels in infected cells. As anticipated, lower ROS levels were recorded in the Δfba mutant (in which KatG expression is not repressed) compared to the wild-type situation. DCFDA levels were also visualized using fluorescence microscopy. The data obtained further support the link between accumulation of ROS and a pro-inflammatory response.

These new results were added in the revised manuscript :

Lines 299-308 « *We next compared the amount of ROS in WT-infected cells to that in Δfba -infected cells, 10 h and 24 h after infection. For this, we used the DCFDA Cellular ROS Detection Assay Kit (Abcam, Cambridge, UK), essentially as described by the manufacturer. The ROS content was approximately 20% lower with the Δfba mutant compared to wild-type, at both time points (**Fig. S10A**). As positive control, non-infected cells were stimulated with $5\mu\text{g mL}^{-1}$ of LPS-EK Standard. LPS stimulation provoked an increase of ROS production that was up to 1.5-fold higher than that in WT-infected cells, at both time-points tested. DCFDA levels were also visualized using fluorescence microscopy. The percentage of fluorescent cells was significantly higher in WT-infected cells (33%) or LPS-stimulated NI cells (63.5%) than in Δfba -infected cells (8.5%) (**Fig. S10B**). Altogether, these assays further supported a direct correlation between the FBA-mediated repression of KatG expression and cellular ROS production”.*

These data are shown in a new figure (**Fig. S10**).

Reviewer 2

The authors have done a very nice job of addressing most of my concerns and they have significantly strengthened the evidence that FBA plays a regulatory role in Francisella. I have one additional comment relating to some of the new data they provide.

Comment

1. The experiments that show direct binding of FBA to the katG and rpoA promoters are a nice addition (Fig. 7B) but appear to lack a specificity control. This is important as it looks as though these electrophoretic mobility shift assays were performed in the absence of an excess of non-specific

competitor DNA (which is normally included at a 100-1000-fold excess over the labelled probe). The authors should simply repeat their current assay using a labelled promoter that is not regulated by FBA as a negative control.

Answer: As recommended by the reviewer, we repeated our current assay using a labelled promoter that is not regulated by FBA as a negative control. For this, we used the promoter region of *uvrD* gene. As expected, no specific shifted bands were observed with this promoter region in presence of purified FBA-HA.

This result was added in the text « *As negative control (right panel), EMSA assays were performed with DIG-labeled *uvrD* promoter region (*p_{uvrD}*, 188 bp). As expected, no specific shifted bands were observed with this promoter region in presence of purified FBA-HA*” and in Figure 7 (Fig. 7B, right panel).